# Technical Note: Unsupervised classification of ozone profiles in UKESM1

Fouzia Fahrin[1,2], Daniel C. Jones[3], Yan Wu[2], James Keeble[4,5], and Alexander T. Archibald[4,5]

[1]Department of Geological and Atmospheric Sciences, Iowa State University, USA
[2]Department of Mathematical Sciences, Georgia Southern University, USA
[3]British Antarctic Survey, NERC, UKRI, Cambridge, UK
[4]Department of Chemistry, University of Cambridge, Cambridge, UK
[5]National Centre for Atmospheric Science (NCAS), University of Cambridge, Cambridge, UK

**Correspondence:** Fouzia Fahrin (ffahrin@iastate.edu)

**Abstract.** The vertical distribution of ozone in the atmosphere, which features complex spatial and temporal variability set by a balance of production, loss, and advection, is relevant for both surface air pollution and climate via its role in radiative forcing. At present, the way in which regions of coherent ozone structure are defined relies on somewhat arbitrarily drawn boundaries. Here we consider a more general, data-driven method for defining coherent regimes of ozone structure; we apply an unsupervised classification technique called Gaussian Mixture Modelling (GMM), which represents the underlying distribution of ozone profiles as a linear combination of multi-dimensional Gaussian functions. In doing so, GMM identifies coherent groups or sub-populations of the ozone profile distribution. As a proof-of-concept study, we apply GMM to ozone profiles from three subsets of the UKESM1 coupled climate model runs carried out for CMIP6: specifically, the seasonal mean of a historical subset (2009-2014) and two subsets from two different future climate projections (i.e., SSP1-2.6 & SSP5-8.5). Despite not being given any spatiotemporal information, GMM identifies several spatially coherent regions of ozone structure. Using a combination of statistical guidance and post-hoc judgment, we select a six-class representation of global ozone, consisting of two tropical classes and four mid-to-high latitude classes. The tropical classes feature a relatively high-altitude tropopause, while the higher-latitude classes feature a lower-altitude tropopause and low values of tropospheric ozone, as expected based on broad patterns observed in the atmosphere. Both of the future projections feature lower tropospheric ozone concentrations than the historical benchmark, with signatures of ozone hole recovery. We find that the area occupied by the tropical classes is expanded in both future projections, which are most prominent during austral summer. Our results suggest that GMM may be a useful method for identifying coherent ozone regimes, particularly in the context of model analysis.

## 1 Introduction

Earth's atmospheric ozone distribution is a topic of interest because of its effect on climate and its role in protecting surface-dwelling organisms from harmful ultraviolet radiation (Newman and Todara, 2003; Monks et al., 2015). The distribution of ozone varies both vertically and horizontally. Nearly 90% of ozone is found in the stratosphere, the layer of the atmosphere between 10-50 km, while 10% is found in the troposphere, the atmospheric layer extending from the surface to 10 km. Strato-

spheric ozone protects surface-dwelling life by reducing the number of high energy photons reaching the surface, which would otherwise lead to high occurrences of skin cancer, cataracts, and impaired immune systems (Newman and Todara, 2003; Monks et al., 2009). In contrast, near-surface tropospheric ozone poses a threat to human health as it is a pollutant (Monks et al., 2015).

The spatial variation in ozone is driven by complex atmospheric processes. Unlike many of the important trace gas species studied in the atmosphere, ozone is not directly emitted from natural or anthropogenic sources. Instead, atmospheric ozone concentrations are controlled by chemical, radiative, and dynamical processes that operate on a range of timescales. Adding further complication is the fact that these processes vary significantly with altitude. In the stratosphere, gas phase photochemical reactions involving oxygen produce ozone (Chapman, 1930), while it is destroyed through reactions involving chlorine, nitrogen, hydrogen, and bromine radical species (Bates and Nicolet, 1950; Crutzen, 1970; Johnston, 1971; Molina and Rowland, 1974; Cicerone et al., 1974). In contrast, tropospheric ozone is produced through photochemical oxidation of ozone precursors such as carbon monoxide (CO), methane ($CH_4$) and non-methane volatile organic compounds (NMVOCs) in the presence of nitrogen oxides (NO and $NO_2$). Similarly, transport processes differ between the stratosphere and troposphere. Because of these different processes, understanding patterns in the vertical distribution of ozone remains a challenge (Monks et al., 2015). These ozone precursors can be transported far downwind from their source locations (Chameides et al., 1992; Monks et al., 2009).

Not only are there significant differences in the processes controlling local ozone mixing ratios at different altitudes, but these processes respond differently to changes in atmospheric composition and global climate. Past changes in anthropogenic emissions, biomass burning, and lightning have all contributed to increased emissions of ozone precursors and increased tropospheric ozone (Griffiths et al., 2021; Jaffe and Wigder, 2012; Monks et al., 2015; Laban et al., 2018). In contrast, emissions of halogenated ozone-depleting substances (ODS) at the end of the 20th century led to significant decreases in stratospheric ozone concentrations and the formation of the ozone hole (Keeble et al., 2021). Future projections of ozone concentrations are dependent on assumptions made about greenhouse gas, ozone precursor, and halogenated ODS emissions, and these changes may work against each other. For example, stratospheric ozone mixing ratios are expected to increase in the coming decades as ODS levels decline. However, an acceleration of the Brewer-Dobson circulation (BDC) associated with increasing greenhouse gas concentrations may lead to reductions in lower tropical stratospheric ozone mixing ratios (Eyring et al., 2013; Meul et al., 2016; Keeble et al., 2017), while increasing the transport of ozone into the mid-latitudes troposphere. Because of these complex interactions, understanding future changes to the vertical distribution of ozone requires simulations performed by complex models (Banerjee et al., 2016; Meul et al., 2018).

Because of this complexity, chemistry-climate and Earth system models are often used to explore changes in atmospheric ozone. A key component in this evaluation is the comparison of ozone derived from different models and/or from different scenarios in the same model (Griffiths et al., 2021; Keeble et al., 2021). Often this is done at the global scale, but if regional comparisons are made, this is often done by averaging ozone profiles over set latitude ranges. However, owing to the complex, spatially heterogeneous processes controlling the distribution of ozone described above, this is a poor method for identifying regions with similar profiles. As climate and ozone mixing ratios change in the future, the boundaries between ozone profiles with similar characteristics might be expected to move. This feature would not be captured by averaging profiles over fixed

latitude ranges. In this work, in order to address this limitation in latitude-based averaging methods, we describe the vertical ozone structure with an unsupervised classification method that groups profiles into classes based on their similarity.

Clustering techniques have already been used in ozone concentration studies for understanding long-term variability. Boleti et al. (2020) have applied a multidimensional clustering technique to understand the long-term trend of ozone. Diab et al. (2004) used a six-cluster analysis which resulted in distinct clusters of "background" and "polluted" with below and above ozone mixing ratios from over 100 ozonesonde profiles launched from a subtropical Southern Hemisphere Additional Ozonesondes (SHADOZ) (Thompson et al., 2003) site, Irene, South Africa. Jensen et al. (2012) performed a cluster analysis named self-

organizing maps (SOM) (Kohonen, 2012) on over 900 tropical ozonesonde profiles. Their findings with four-cluster results were similar to Diab et al. (2004). Both studies showed that the seasonal influences of biomass burning and convection dominate ozone variability. Stauffer et al. (2016) documented the influence of meteorological conditions on the shape of the ozone profile from the troposphere to the lower stratosphere by applying the SOM clustering technique to ozonesonde data from specific northern hemisphere midlatitude geographical regions. Later they expanded the study for global ozonesonde sites to show the

variation of ozone profiles cluster for various regions and how they vary based on meteorology and chemistry depending on latitudes (Stauffer et al., 2018).

In our study, we adopt a Gaussian Mixture Modelling (GMM) approach, an automated, robust, and standardized unsupervised classification technique that has previously been applied to ocean structure and dynamics (Bishop, 2006; Maze et al., 2017; Jones et al., 2019; Sonnewald et al., 2019; Rosso et al., 2020). GMM does not use any latitude or longitude information

to identify similar profiles and cluster them together, which makes it more general than a latitude-based averaging method. In section 2, we describe the method adopted in the study and the data set used in the study. In section 3, we present the results of the GMM-based clustering analysis. Finally, we end with a brief discussion in section 4 and conclusions in section 5.

## 2  Methods and data

Our approach is based on Gaussian Mixture Modelling (GMM), which is a type of unsupervised classification method. We

want to model the vertical ozone structure, i.e., to understand how we can identify different ozone profile types in a dataset. To do so, we analyze the diversity of vertical ozone profiles by way of identification of recurrent patterns throughout the collection of profiles using unsupervised learning.

### 2.1  UKESM1 Experiment Selection

The UK Earth System Model 1 (UKESM1, https://ukesm.ac.uk/) is a coupled climate model with a well-resolved strato-

sphere, tropospheric-stratospheric chemistry, ocean-atmosphere carbon and aerosol coupling, and terrestrial biogeochemistry (Sellar et al., 2019). The model has a horizontal resolution of $1.25^0$ latitude by $1.875^0$ longitude, with 85 vertical levels on a terrain-following hybrid height coordinate and a model top at 85 km ( 0.004 hPa). UKESM1's complex physical-biogeochemical coupling and its realistic representation of historical ozone structure and trends make it a suitable choice for our study (Keeble et al., 2021). Using the Pangeo platform, we selected annual mean ozone profile data from three different

UKESM1 experiments (Abernathey et al., 2021). We chose seasonal means to include seasonal variations in ozone structure. Changes in ozone precursor emissions have an effect on future tropospheric ozone concentrations; reductions in precursor emissions drive ozone decreases in shared socioeconomic pathways (SSPs) (Griffiths et al., 2021). To explore the effect of emissions on the class properties, we used ozone data from three different experiments:

- **Historical**: Seasonal means covering the years 2009-2014.

- **SSP1-2.6**: Seasonal means covering the years 2095-2100 (strong emission reductions).

- **SSP5-8.5**: Seasonal means covering the years 2095-2100 (no emission reductions).

Here each simulation year contains 110591 seasonal mean profiles.

In order to create a training dataset for the GMM algorithm, we combined data from all three of the above experiments. Essentially, we trained the GMM in such a way that it "sees" structures from all three experiments and is thereby better able to represent the full range of possible structures, i.e., the training process is not biased towards one particular experiment. Using the trained GMM, we labeled the full dataset of ozone profiles from all three experiments. We then used the fully labeled dataset to look for differences in structure among the historical, SSP1-2.6, and SSP5-8.5 experiments.

At present, standard implementations of GMM cannot handle missing values. So in this context, one has to select a subset of the ozone profiles that feature values on every selected standard pressure level. We discarded any profiles with NaN values. As such, we only worked with profiles with values over the entire pressure range, from 1-850 hPa. This means that much of our analysis takes place over the ocean and only partially covers land-based areas, i.e., out of necessity, we omit grid cells with surface pressures lower than 850 hPa due to topography.

## 2.2 Gaussian mixture modelling

Gaussian Mixture Modelling (GMM), a machine learning method, uses a probabilistic approach for describing and classifying data by representing the underlying data distribution using a linear combination of multi-dimensional Gaussian functions (McLachlan and Basford, 1988). By using a sufficient number of Gaussians, any continuous density field can be approximated to arbitrary accuracy. This allows us to identify and model the typical vertical structure represented in the collection of profiles.

Although GMM has been used in several oceanographic studies to date (Maze et al., 2017; Jones et al., 2019; Sonnewald et al., 2019; Houghton and Wilson, 2020; Sonnewald et al., 2020; Rosso et al., 2020; Desbruyères et al., 2021; Boehme and Rosso, 2021), to our knowledge, our application is novel in the field of atmospheric chemistry. One unique aspect of this approach is that we do not use any geographical information about the profiles to identify groups of similar profiles. Specifically, we withhold latitude, longitude, and time information from the unsupervised classification algorithm; it only sees the values of the ozone concentration on each standard pressure level. The motivation behind withholding the geographical information is that we want the algorithm to cluster the profiles without spatial information, and still, the class structure can explain most of the information when plotted spatially.

The core foundation of a GMM, as described in Bishop (2006), is that any Probability Density Function (PDF) can be described as closely as desired with a model of weighted sums of Gaussian PDFs:

$$p(x) = \sum_{k=1}^{K} \lambda_k N(x|\mu_k, \Sigma_k) \tag{1}$$

which is called a mixture of Gaussians. Each Gaussian density $N(x|\mu_k, \Sigma_k)$, a multidimensional normal probability density function (PDF), is called a component of the mixture and has its own mean $\mu_k$ and covariance $\Sigma_k$. Where $x$ is a single profile taken from the complete array $X$.

We use an expectation-maximization algorithm (Appendix B) to find the maximum likelihood solution for the model, which is effectively "training" the GMM to represent the underlying structure of the ozone data as represented in abstract principal component space (section 2.3).

## 2.3 Dimension reduction

The abstract "feature space" in which we perform the clustering is relatively high-dimensional; ozone is defined on 19 standard pressure levels in our dataset. Because GMM becomes less efficient for high-dimensional problems, we apply a dimension reduction scheme to reduce the computational expense of the training step. A large number of dimensions in the problem fundamentally translates into a large number of parameters to be determined in the Gaussian covariance matrices. Here we used Principal Component Analysis (PCA), a dimension reduction method that is often used to reduce the dimension of large data sets by transforming a large set of variables into a smaller set that still retains an acceptable percentage of the variability.

As a first prepossessing step, we standardize the ozone values on each pressure level. Since the ozone values on each pressure level are standardized independently, "small" variations in ozone on levels with low variability can have roughly the same effect as "large" variations in ozone on levels with high variability. This ensures that the structure seen by GMM is not just dominated by the pressure levels on which the variability is high. This prepossessing step also helps to speed up the algorithm (Jaadi, 2019).

In the last step of PCA, we express each ozone profile as a linear combination of eigenfunctions using the following equation for $x(z)$:

$$x(z) = \sum_{j=1}^{d} P(z,j) y(j) \tag{2}$$

where $z$ is the pressure level, $d$ is the total number of PCs (index $j$), and $P(z,j)$ is the transformation matrix between pressure space and PC space. $P \in \mathbb{R}^{D \times d}$ and $y \in \mathbb{R}^{d \times N}$ with $d \leq D$. The first row of $P$ contains profiles maximizing the structural variance throughout the collection of profiles. Thus, if we choose $d \leq D$, we can reduce the number of dimensions of the data set x while preserving most of its structure. This creates a new space where the $N$ profiles are not defined with $D$ vertical level values (the $x$ array) but with only $d$ values ($y$ array). The transition between one space to the other is done

through the matrix $P$ containing the definition of the new dimensions in the original ones ($d$ vertical profiles of D levels, the eigenvectors of the covariance matrix $x^T x$) (Figure A1).

We find that with 10 PCs, this transformation captures 99% of the variance in the vertical structure, which appropriately reduces the number of dimensions we need to describe the profile structure from UKESM1, that is, from 19 pressure levels to 10 PCs. A reduction to an even smaller number of PCs is possible at the expense of losing more of the variability in the original dataset.

## 2.4   Selection of the number of classes

We used a random sampling technique to select a subset to perform *Bayesian Information Criterion* (BIC) to find the appropriate $K$ for classes. We refer the readers to Appendix C for details of the BIC. The reason for random sampling is to test the sensitivity of our results to the sample selection process. Under random sampling, each observation of the data set subset has an equal opportunity to be chosen as a part of the sampling process. Note that this sampling is not related to unbiased spatial sampling.

In our application, for each potential value of $K$, we chose 20 different sets of 1000 random samples from the full dataset of 442,364 profiles. This sampling approach allowed us to estimate the mean and standard deviation of BIC at each $K$. We used the same random seed each time, so there is no variability associated with the random initial guesses for the cluster centers. The mean BIC curve appears to flatten after $K = 6$, indicating a point of diminishing returns for increasing $K$ (Figure 1). The overfitting penalty term starts to dominate for $K > 12$, indicating an upper bound for the number of classes.

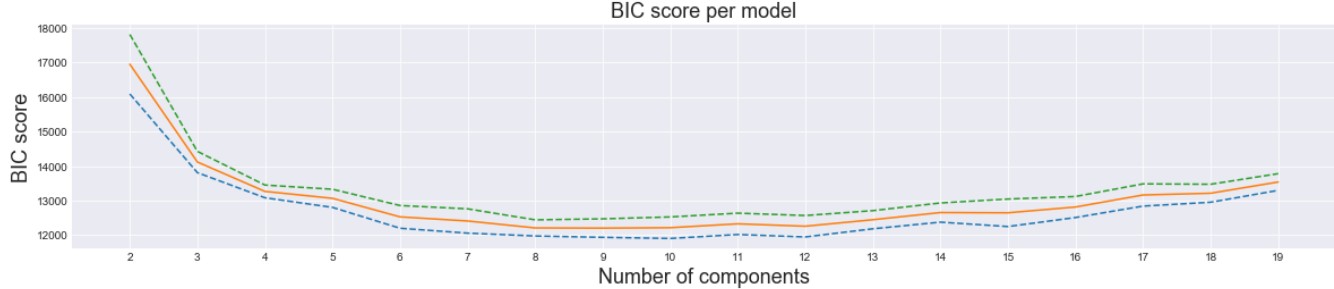

**Figure 1.** BIC Score versus the specified number of classes K for UKESM1 data. The solid line is the mean BIC value, and the dashed lines represent one standard deviation on either side of the mean.

## 3   Classification of UKESM ozone profiles

### 3.1   Classification of ozone profiles from different experiments

In this section, we analyze the general vertical structure of ozone data from the UKESM simulations that represent a chosen historical period (2009-2014) and two future projection datasets, as mentioned in section 2.1. Our results are not especially

sensitive to the choice of any particular dataset from the 3 experiments since we train the GMM using all the profiles from each period from a variety of atmospheric ozone states. The classes are sorted by mean latitude for ease of interpretation.

Proceeding from south to north: classes 1 and 2 are high-latitude Southern Ocean classes with similar mean profiles but different variability structures as measured by the standard deviation curves (Figure 2). They both feature relatively low-altitude and gentle tropopauses, as indicated by the slope of the ozone curves. Class 1 has the lowest ozone value at 850hPa (Table 1); it has a significant amount of variability in the middle stratosphere, which is associated with the ozone hole (Wargan et al., 2020), which has the largest effect on class 1, based on its intensification with respect to the season at high southern latitudes. The mean posterior probability, which in the context of a given statistical model is a measure of the algorithm's confidence in its assignment, is somewhat lower for class 2 than for class 1, indicating that there is some ambiguity associated with the assignments into class 2, which may be somewhat of a boundary or transition class between the high southern latitudes and the tropics. Note that high posterior probabilities do not necessarily indicate that the particular GMM is the best fit to the data, only that the selected GMM is confident in its assignment as measured by the uncertainty. Class 2 is also highly variable throughout the upper troposphere and tropopause but not as class 1. Notably, all of the high-latitude Southern Hemispheric classes feature relatively low lower tropospheric ozone values with small variability - they are relatively "clean" in terms of surface ozone pollution (Table 1).

Classes 3 and 4 are tropical classes, with higher lower tropospheric ozone concentrations and a higher-altitude tropopause compared with the Southern Hemispheric classes (Figure 2). Class 3 and class 4 share similar kinds of structures from the lower troposphere to the upper stratosphere. Class 4 features higher lower tropospheric ozone and higher variability than class 3. Finally, classes 5 and 6 are northern hemispheric classes with high lower tropospheric ozone concentrations and large variability from the tropopause to the stratosphere. The higher lower tropospheric values result from greater surface pollutants in classes 4, 5, and 6, including the associated ozone precursor emissions, which tend to be concentrated in the Northern Hemisphere due to anthropogenic emissions (Monks et al., 2009, 2015).

Progressing from south to north, we see that the altitude of the maximum ozone concentration generally increases in height from the high-latitude southern hemisphere to the tropics and then decreases in height from the tropics to the high-latitude northern hemisphere (Figure 2). This structure is consistent with observations and is enforced by the meridional Brewer-Dobson circulation (Butchart, 2014), which is associated with upwelling in the tropics and downwelling in the extratropics, somewhat favoring the southern hemisphere (Butchart, 2014; Li and Thompson, 2013; Newman and Todara, 2003; Weber et al., 2011). The imprint of this circulation pattern is a low-altitude tropopause at the poles and a higher-altitude tropopause at the equator.

### 3.1.1 Classification of ozone profiles from historical experiment

The vertical ozone structure change pattern is complex following seasonal variation. To examine how the spatial pattern of the classes changes with seasons, global mean ozone concentrations are plotted according to seasonal categorization. The label map indicates the geographic distribution of the classes during 2009-2014 (Figure 3). Notably, although the GMM algorithm was not given any latitude or longitude information, it was nevertheless able to identify spatially coherent groups. The tropical

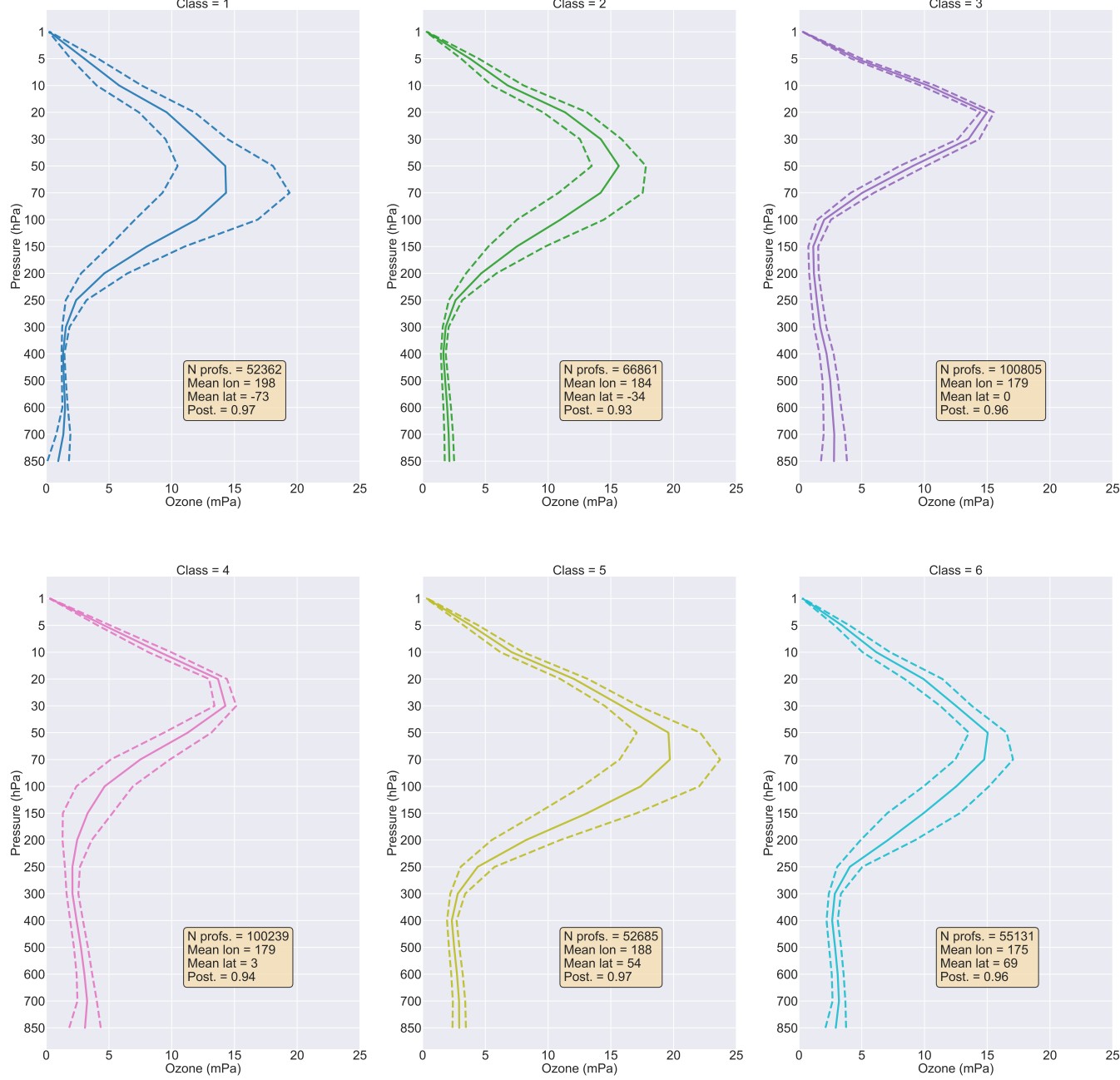

**Figure 2.** Ozone concentration statistics of UK Earth System model data for the whole data set, separated by class, as a function of pressure, sorted by latitude. Shown are the mean (solid lines) and the mean plus or minus one standard deviation (dashed line) for all profiles in the indicated class. Also shown are the number of profiles in each class and the class mean values for longitude, latitude, and posterior probability.

# UKESM1 Historical Label Map

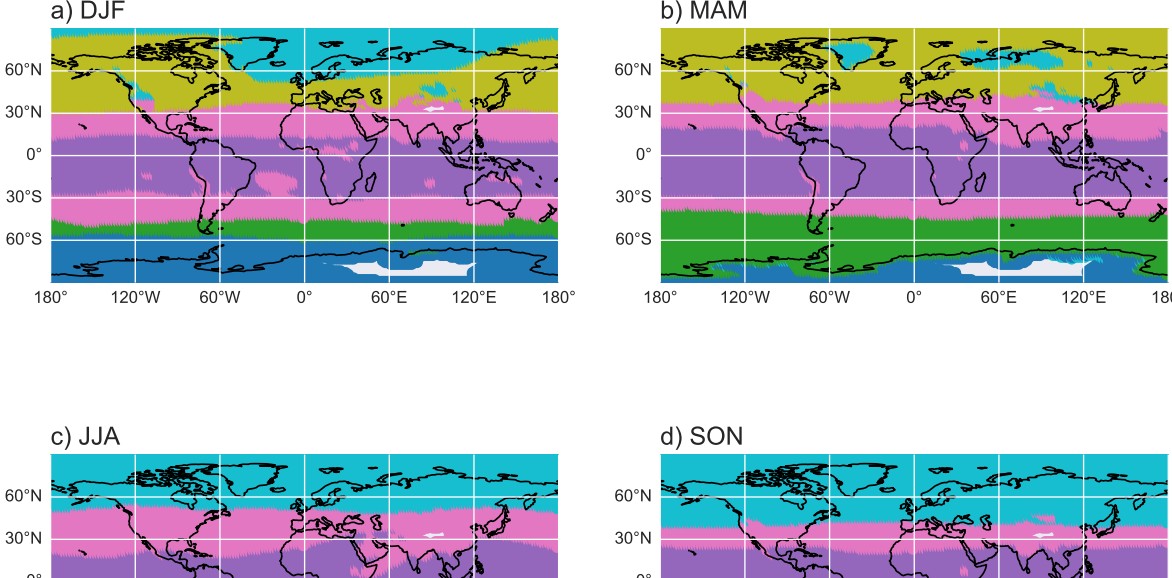

**Figure 3.** Map of profiles color-coded with the class they have been attributed to for model historical data (seasonal mean profiles covering 2009-2014 at each model grid cell) at 850 hPa.

classes are largely organized in roughly zonal bands in each season, with some exceptions (e.g., June-August), where the southern hemispheric and tropical classes shift to the northernmost position (Figure 3c).

Figure 2 shows from the tropopause to the stratosphere, the high latitude and polar classes feature a relatively large standard deviation, especially in the lower and mid stratosphere, suggesting that these classes consist of a wide variety of profiles. These high-latitude classes are more sensitive to seasonal change than tropical classes. For example, from Figure 3, Class 1 (with the largest standard deviation at stratosphere) extends up to 40° S during December-February (DJF), then recedes southward during March-May (MAM), and it starts migrating northward again during June-August (JJA). It reaches its northernmost position again in September-November (SON). The stratospheric ozone ($14.33 \pm 5.07$ mPa) suggests that depending on the strength Antarctic ozone hole, this value varies with the season, and during SON, the region covered by class 1 contains the lowest amount of stratospheric ozone (Table D4). On the other hand, class 2 starts shifting northward during MAM and reaches

its northernmost position during JJA. Since the southern polar vortex is much stronger, it prevents the mixing of classes 1 and 2 during the southern fall and winter seasons. The tropical classes are less variable except for DJF and JJA. The tropical classes shift to the southernmost position during the former and expand up to around 50° N during the latter. Class 3 expands most northward during MAM. The widening trends based on seasonality implies that the tropical broadening in SH is mainly due to the Antarctic ozone hole, which causes the largest radiative cooling effect in the lower stratosphere during DJF (Palmeiro et al., 2014). Increasing black carbon and tropospheric ozone are considered as major forcing for NH tropical class widening on a longer time scale during JJA (Allen et al., 2012). However, these two forcings together have the largest warming effect in the NH extratropics (Hu et al., 2018). Studies showed that the shallow branch (located in the lowermost stratosphere with upwelling in the tropics and downwelling in the subtropics) of tropical upwelling is much stronger toward the summer hemisphere during DJF than JJA (Palmeiro et al., 2014). The deep branch with upwelling in the upper stratosphere in the tropics and downwelling in the middle and high latitudes also show a similar seasonal cycle with downwelling extended to the polar latitudes in the stratosphere (Seviour et al., 2012; Palmeiro et al., 2014). The differentiation between twp branches are based on different forcing, planetary-scale wave forcing act on the shallow branch, and in the deep branch, the upwelling is associated with GHG increase (Palmeiro et al., 2014). However, the investigation of seasonal change of tropical upwelling in shallow and deep branches is beyond the scope of this study.

The northern high-latitude classes are characterized by frequent variability. Spatially, Class 5 is a dominant northern subpolar and polar class during DJF and MAM. For the remainder of the year, class 5, with a very high amount of stratospheric ozone concentration (Fig. 2), is absent, and class 6 dominates the entire region (Fig. 3c & 3d). This tendency suggests us Arctic high-altitude ozone is stronger during northern hemispheric winter and spring and weaker for the rest of the year (Appendix D).

The high lower tropospheric ozone concentration in classes 4 and 5 (Table 1) highlights anthropogenic emissions over those regions and the bulk of biomass burning and wildfire, which occurs primarily near the Arctic Circle, Africa, and some parts of North America (Laban et al., 2018; Jaffe and Wigder, 2012). In the last few decades, wildfires/biomass burning have gained much attention as they have been recognized as the second–largest source of ozone precursor emissions (Monks et al., 2015). Boreal forest fires are a known source of high-surface ozone over North America (Jaffe and Wigder, 2012). Biomass burning in Africa produces a significant amount of ozone precursor. Arctic boreal fire and biomass burning are sources of high ozone precursors over the Northern extratropical and temperate zone (Laban et al., 2018; Monks et al., 2015; Jaffe and Wigder, 2012).

Classes at the northern high latitudes (i.e., class 5) have more stratospheric ozone than those at southern high latitudes (i.e., classes 1 & 2), and this class peaks during DJF and MAM (Table D1 & D2). This indicates that in our study, the northern hemisphere ozone hole is not especially predominant during these months in seasonal mean. However, Dunn et al. (2022) showed that there are some particular years when the polar ozone hole can happen in NH spring. Larger amplitudes of upward propagating planetary waves like Rossby waves can propagate from troposphere to stratosphere with eastward wind, where these waves can perturb stratospheric circulation and reduce the speed of polar night jet (Lee, 2021; Oehrlein et al., 2020; Waugh et al., 2017). In the Northern Hemisphere, the continents' and mountain ranges' layouts accelerate this wave activity more than in the Southern Hemisphere (Lee, 2021; Waugh et al., 2017). Consequently, the Arctic stratospheric vortex is much

## UKESM1 SSP1-2.6 Label Map

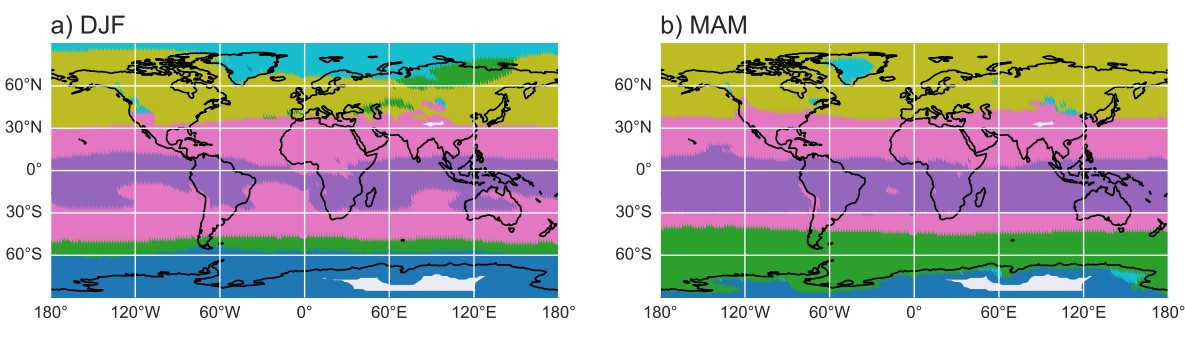

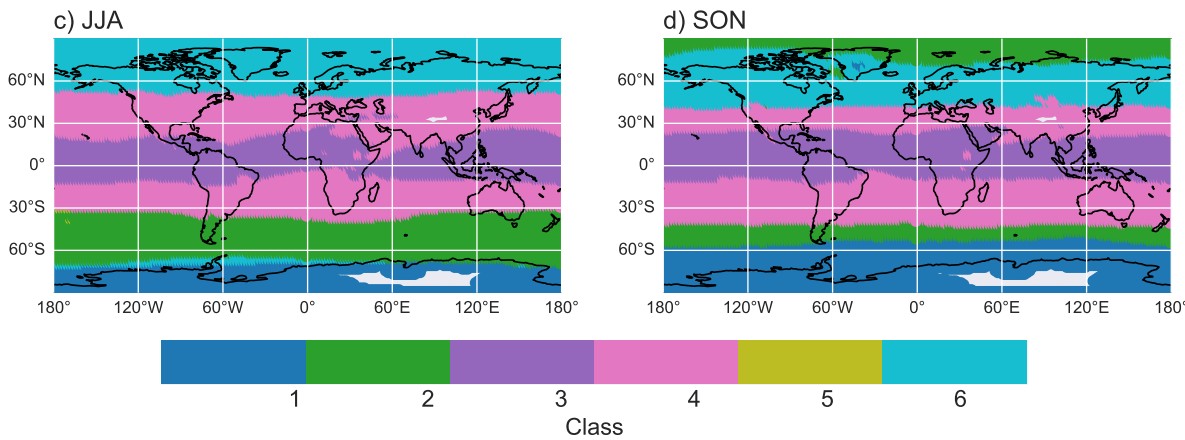

**Figure 4.** Same as figure 3, but for SSP 1.2-6 Label Map covering the year 2095-2100.

weaker and more variable than its Antarctic counterpart, which features larger meanders in the meridional extent. It is for this reason that, unlike the Antarctic, a large ozone hole does not form in the Arctic stratosphere each winter. As the Arctic temperature is higher than the Antarctic, a strong Antarctic vortex allows for the formation of polar stratospheric clouds that catalyze ozone depletion (Waugh et al., 2017; Lee, 2021; Newman and Todara, 2003). This allows redistribution of stratospheric ozone and pulls ozone from the tropics in the Northern Hemisphere (Lee, 2021; Newman and Todara, 2003). The strong polar vortex in the south pole prevents the region from having high stratospheric ozone (Newman and Todara, 2003), especially during the Antarctic spring season.

### 3.2 Classification of ozone profiles in the future climate projections SSP1-2.6 and SSP5-8.5

We examine the distribution and structure of ozone in two chosen future climate projections, namely SSP1-2.6 and SSP5-8.5. SSP1-2.6 is a scenario with strong emission reductions, and SSP5-8.5 is with increased emissions. We chose these two experi-

## UKESM1 SSP5-8.5 Label Map

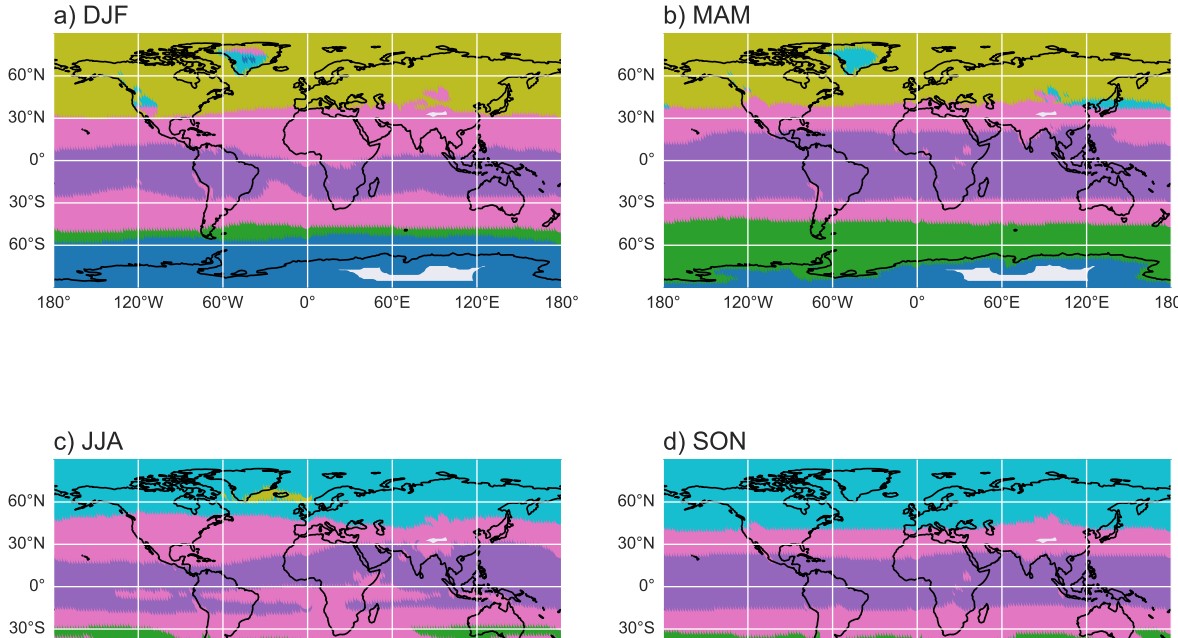

**Figure 5.** Same as figure 3, but for SSP5.8-5 Label Map covering the year 2095-2100.

ments as end-members representing two drastically different future projections. In the SSP1-2.6 case, with reduced emissions of ozone precursors, the total lower tropospheric ozone concentration gets smaller (Table 1). In the SSP5-8.5 case, with increased emissions of ozone precursors, the total lower tropospheric ozone concentration is slightly increased or approximately steady (Table 1).

Classes 1 and 6, in particular, which are affected by the ozone hole because of their geographical location, display a variation in stratospheric ozone (Appendix D) between 2009-2014 and 2095-2100 in both cases in each season but for the southern hemisphere, SON dominates the increase of stratospheric ozone (Table D4), which is a signature of the closing of the ozone hole (Keeble et al., 2021). The maximum concentration is located around 30 hPa in the historical case, which is above the region of maximum ozone depletion. The recovery of the ozone hole also shifts the level of maximum ozone concentration to lower altitudes (higher pressures, i.e., from 30 hPa to 70 hPa) for the southern hemisphere in future projections austral

spring season (Table D4). In the next subsections, we investigate differences in the spatial structure of the two future emissions experiments.

### 3.2.1   Geographical distribution of Ozone profiles in SSP1-2.6

Here we examine the spatial pattern of ozone profiles in SSP1-2.6 in each season over the year 2095-2100. As with the historical experiment, class 1 has the lowest 850 hPa ozone (Table 1), which is consistent with the reduction in surface ozone precursors in this experiment. The maximum value of stratospheric ozone increases under this scenario, which is a signature of the recovery of the ozone hole (Keeble et al., 2021).

Moving northwards, class 2 appears to have a similar structure to its historical counterpart, with higher stratospheric ozone
and considerable variability in the upper troposphere to the middle atmosphere (Figure 2). It is a mid-latitude southern hemispheric class occupying roughly the same total surface area as it did in the historical experiment (Figure 3 & 4). Unlike the historical case, the area occupied by class 3 has decreased during DJF, and also, in other seasons, this class shifts northward and southward, as it was in the historical case (Fig. 4). This suggests strong emissions play a vital role for class 3. Notably, the relative position of class 2 sits next to class 6 during DJF and SON, indicating that these two classes may be difficult to
unambiguously differentiate over these seasons because of their similar structure. Classes 5 and 6 geographic distribution are similar to their historical counterpart, except with reduced lower tropospheric ozone concentrations consistent with continued ozone precursor emissions reductions (Table 1) and increased stratospheric ozone during DJF and MAM (Table D1 & D2). The tropospheric ozone decrease is more significant in the NH than in other scenarios, helping to mitigate climate change and air quality impacts (Table 1) (Keeble et al., 2021).

### 3.2.2   Geographical distribution of ozone profiles in SSP5-8.5

Here, we examine the structure of atmospheric ozone in the 2095-2100 years of the SSP5-8.5 experiment. In this experiment, ozone mixing ratios are generally higher throughout much of the troposphere and upper stratosphere. In the troposphere, the drivers of this increase are complex. Under the assumptions of the SSP5-8.5 scenario, global mean emissions of nitrogen oxides (NOx) and carbon monoxide (CO) are lower in 2095 than in the present day, while global mean emissions of methane (CH4) are
higher (Gidden et al., 2019). However, changes to ozone precursor emissions (including biogenic volatile organic compounds (BVOC) emissions caused by increasing tropospheric temperature) alone do not drive tropospheric ozone changes; the availability of tropospheric water vapor and stratosphere-to-troposphere transport of ozone, which taken together drive increases to tropospheric ozone concentrations (Griffiths et al., 2021; Turnock et al., 2020; Zanis et al., 2022). In the stratosphere, this increase is simpler to understand. Upper stratospheric ozone increases under all SSPs as ozone-depleting substances decrease
but increases more in scenarios that assume larger increases in greenhouse gas emissions due to the resulting CO2-induced cooling of the stratosphere and the impacts this has on gas-phase chemistry (Haigh and Pyle, 1982; Jonsson et al., 2004).

Proceeding from south to north, we see that classes 1 and 2 are similar to their historical counterparts during DJF and MAM, covering a similar proportion of area, albeit with increased stratospheric ozone at the pressure level with maximum concentration during JJA and SON, and it decreases during both DJF and MAM unlike SSP 1-2.6 and historical case (appendix D).

| Class | Hist (mean) | (std) | SSP1-2.6 (mean) | (std) | SSP5-8.5 (mean) | (std) |
|-------|-------------|-------|-----------------|-------|-----------------|-------|
| 1 | 0.990 | 0.890 | 0.900 | 0.790 | 1.040 | 0.880 |
| 2 | 2.150 | 0.380 | 1.980 | 0.330 | 2.160 | 0.450 |
| 3 | 2.990 | 0.950 | 2.210 | 0.700 | 2.560 | 1.080 |
| 4 | 3.360 | 1.290 | 2.620 | 0.920 | 3.190 | 1.330 |
| 5 | 3.190 | 0.440 | 2.310 | 0.260 | 3 | 0.440 |
| 6 | 2.940 | 0.780 | 2.250 | 0.610 | 3.290 | 0.850 |

**Table 1.** Ozone concentration statistics at 850 hPa for the historical, SSP126, and SSP585 experiments, shown in mPa (from Fig. 2 but for each experiment)

Future ozone depletion decrease will lead to ozone concentration increase throughout the atmosphere, and both hemispheric high-latitude upper stratosphere will have the largest changes (Griffiths et al., 2021). However, an increasing amount of greenhouse gas emission will yield a more complex pattern of ozone changes, which will lead to a possible strengthening of the Brewer-Dobson circulation to an increase in net stratospheric influx, and high tropospheric ozone in the Southern Hemisphere class is the result of circulation changes (Young et al., 2013; Monks et al., 2015; Butchart, 2014; Griffiths et al., 2021; Lu et al., 310 2019).

The tropical classes (i.e., 3 and 4) are similar to those seen in the historical case, except for JJA. During JJA, class 3 is more sparse in the southern hemisphere. Interestingly class 5 starts showing up in the southern polar region during JJA (Figure 5). This experiment is associated with an enhanced amount of ozone mixing ratio, which causes the polar vortex to weaken. As a result, during SH winter, a huge amount of stratospheric ozone sits next to class 1. Finally, class 6 remains a large-scale 315 Northern Hemispheric polar class during JJA and SON, although class 6 has increased lower tropospheric ozone concentrations relative to SSP1-2.6, in part due to continued precursor emissions. In response to tropospheric warming driven by greenhouse gas in SSP5-8.5, the subtropical tropospheric jets intensify, while the contribution of gravity waves increases in the middle stratosphere (Palmeiro et al., 2014). As a result, stratospheric ozone increases in high latitude classes (Table D).

The oceans are major sinks of tropospheric ozone at the surface, and there are few direct sources of ozone precursors present 320 over the ocean (Archibald et al., 2020a, b). Advection of emission-driven ozone production over the land or an increase of ozone transport from the stratosphere is responsible for ozone increase for the profiles that are covering the ocean (e.g., class 3, which covers the majority of the oceanic region in the tropics) (Archibald et al., 2020a, b).

## 4   Discussion

The distribution of ozone in the atmosphere is relevant for both climate and human health. Recently, researchers have em-
ployed a number of approaches for identifying different "profile types" in both observational and numerical model data, going beyond a basic latitudinal-averaging framework for comparison. These methods complement each other and add to existing

| Season | Historical | SSP1-2.6 | SSP5-8.5 |
|--------|-----------|----------|----------|
| DJF | 64.80 | 66.90 | 66.30 |
| MAM | 65.50 | 66.00 | 66.70 |
| JJA | 65.70 | 67.60 | 65.60 |
| SON | 65.00 | 67.60 | 62.60 |

**Table 2.** Relative area coverage by tropical classes (3+4) combined regions during each season, shown in percentages.

expertise-driven classification approaches. Here we aimed to add to the atmospheric analysis toolbox using unsupervised classification, which is a type of machine learning that identifies patterns and structures in unlabeled datasets. We based our profile classification scheme on Gaussian Mixture Modelling (GMM), which attempts to represent the ozone profiles as represented in
an abstract principal component space using a linear combination of Gaussian functions. We applied GMM to a collection of seasonal mean ozone profiles taken from a set of UKESM1 simulations. Specifically, we used GMM to classify profiles from a historical experiment and two future climate experiments, namely SSP1-2.6 and SSP5-8.5. We used GMM as a "hypothesis generation tool", generating ideas for further exploration and analysis (Kaiser et al., 2022). Note that the detailed exploration of this hypothesis is beyond the scope of this technical note; further analysis of the ideas presented here would be a welcome
addition to the literature. The spatial extent and seasonal variability of the classes reflect the integrated effect of a number of different processes and timescales, so they should be interpreted within that context. Nevertheless, GMM was indeed able to identify spatially coherent profile types and track their variability over time, highlighting the ability of GMM to identify and follow structures.

Even though the GMM algorithm was not supplied with the latitudes or longitudes of the profiles, the classes nevertheless
vary structurally with latitude, as expected. For example, we find two tropical classes (classes 3 and 4) with elevated tropopause heights and two polar classes (classes 1 and 6) with lower tropopause heights, broadly consistent with the imprint of the Brewer-Dobson circulation.

The spatial distributions of the classes generally vary with the season. In the historical UKESM1 experiment, we see that the tropical classes (classes 3 and 4) shift in mean latitude towards the summer pole, i.e., southwards in DJF and northwards
in JJA. The subpolar and polar classes in the Northern Hemisphere (classes 5 and 6) vary drastically, with class 5 disappearing entirely in summer and autumn. This may reflect larger variability in the profile structure seen in autumn and winter. In the Southern Hemisphere, the southernmost class (class 1) usually covers Antarctica, except in the autumn and winter (MAM, JJA) when class 2 covers a larger area. We see similar patterns in SSP1-2.6 and SSP5-8.5, with the notable exception of the appearance of class 5, ostensibly a Northern Hemispheric class, in the wintertime Southern Hemisphere of SSP5-8.5. This result
highlights that the classes are not inextricably tied to a particular latitude band: they may appear wherever similar structures exist. The appearance of class 5 here suggests a shift in ozone distribution large enough that it disrupts the classification scheme, highlighting an area for further study.

Our results for SSP1-2.6 are broadly consistent with the tropical broadening hypothesis in that the spatial extent of the tropical classes (classes 3 and 4) increases between the historical case and SSP1-2.6 across all seasons. We also saw increases under SSP5-8.5, with the possible exception of SON. In the projections of future climate considered here, both Hemispheric high latitudes show large variations in stratospheric ozone. These changes in the ozone concentration for high-latitude classes (i.e., classes 1 and 6) in future projections show the potential changes due to changes in precursor emissions and changes in ozone advection. Southern Hemispheric tropospheric ozone levels are generally low for all three cases considered here. There are larger fluctuations in the lower troposphere at high latitudes of the Northern Hemisphere (class 6), which could be related to differences in precursor transport and chemistry from lower latitudes.

This study focuses on model analysis. When working with model data, we typically have access to fairly uniform spatial and temporal ozone coverage, at least in parts of the atmosphere with a full range of pressures from 850-1 hPa. This coverage allows us to train our mixture model in a way that is relatively unbiased with respect to location and time. The trained mixture model is thus able to identify coherent regimes with similar patterns of vertical variability in a way that is more general than drawing somewhat arbitrary latitude-longitude boxes. Because we can train the mixture model using data from a variety of times and experiments, it is possible to train a GMM that can, in principle, represent the full range of data structures found within a selected ensemble and track how those structures evolve over time. Although we did not attempt to do so here, it should be possible to use GMM for inter-model comparison, allowing for the structures and differences in structures to be derived directly from model data.

Although our study focused on model analysis, it is possible to apply GMM to observed ozone profiles as well. At present, ozone observations are biased towards a few specific locations where long-term monitoring has taken place. Training a GMM on this data would necessarily bias the classes towards particular locations and times, making direct comparisons between models and observations difficult. One possible solution would be to train a GMM on model data and then apply it to observations, although any systematic biases would have to be treated carefully during the data cleaning and prepossessing steps. In terms of working towards a more optimized ozone observing system, it may be useful to use GMM and similar classification methods to identify which regions feature coherent variability.

## 5 Conclusions

In this study, we applied Gaussian Mixture Modelling (GMM), an unsupervised classification method, to ozone profiles from the UKESM1 coupled climate model in order to robustly and objectively identify coherent sets of ozone profile types. Our motive was to investigate the ozone structure using a limited number of classes. We used Principal Component Analysis (PCA) to reduce the computational complexity of the problem, increasing the computational efficiency at the expense of only 1% of the variability in the dataset. We used a statistical approach (i.e., BIC) and post-hoc expert judgment to inform our choice of the number of classes, settling on a six-class representation of the ozone profiles. This six-class system included two tropical classes and four mid-to-high latitude classes. We found that, although the GMM algorithm was not given any spatiotemporal information, it was able to identify a set of spatially coherent regions of ozone structure. We trained the GMM

using data from all three model cases in order to expose it to the full range of profile types in our classification problem. We compared lower troposphere and maximum ozone concentrations for three model cases and their spatial extents. Higher concentrations of stratospheric ozone in classes 1 and 6 in both of the future projection cases indicate a seasonal decrease in ozone depletion and possible ozone hole recovery, which results in a decrease in tropopause height based on seasons (Appendix D). The modelled lower tropospheric ozone is higher in the Northern Hemisphere (NH) and relatively lower in the Southern Hemisphere (SH) (Table 1). Notably, the spatial area occupied by the tropical classes increased in both future projections based on seasonality relative to the historical benchmark, in consistency with the tropical broadening hypothesis, i.e., the expected expansion of tropical upwelling (Table 2). GMM can be applied to identify data-derived regions of coherent ozone structure and may therefore be useful for model-model comparison or model-data comparison.


# Appendix A: Principal Component Analysis (PCA)


The principal component analysis shown in figure A1 is adopted for dimensionality reduction in this work. The figure shows the eigenfunctions. These eigenfunctions came from the eigenvalues and corresponding eigenvectors of the covariance matrix to find the directions along which the variability is the largest.

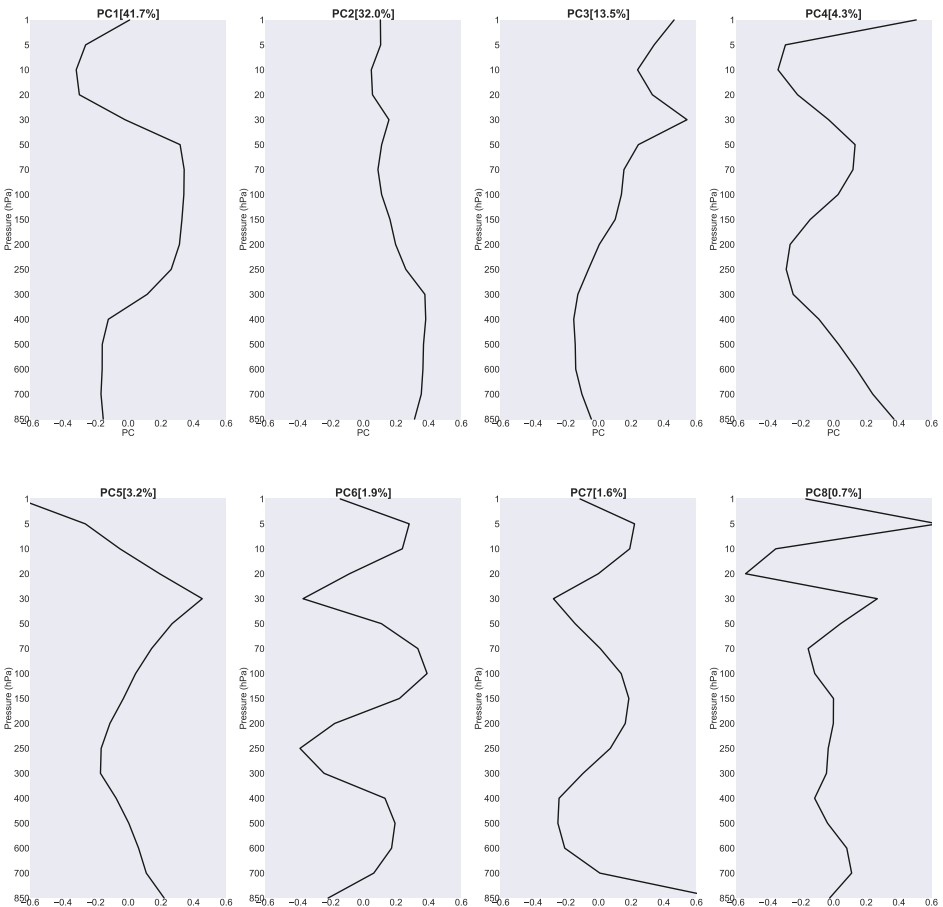

**Figure A1.** Principal components (PCs) with percent variance statistically explained by each PC is shown (in parenthesis).

# Appendix B: GMM details

For details of the GMM classification algorithm, we refer the readers to Bishop (2006). The classification algorithm is adopted from Bishop (2006); Maze et al. (2017)

## B1 Probability density function of profiles

The key ingredient of GMM: a multidimensional normal probability density function (PDF) with mean $\mu$ and covariance $\Sigma$:

$$N(x|\mu,\Sigma) = \frac{1}{\sqrt{(2\pi)^D}|\Sigma|^{1/2}} \exp\left(-\frac{1}{2}(x-\mu)^T\Sigma^{-1}(x-\mu)\right) \tag{B1}$$


In this study, $x \in \mathbb{R}^{D\times 1}$ is a profile of the $\mathbf{X} \in \mathbb{R}^{D\times N}$ collection, $\mu_k$ is a D-dimensional mean vector and $\mu_k \in \mathbb{R}^{D\times 1}$, $\Sigma \in \mathbb{R}^{D\times D}$ a covariance matrix and $|\Sigma|$ is the determinant.

In other words, the array $\mathbf{X}$ is the data set we want to analyze; it is made of N vertical profiles (as columns) of D pressure levels (as rows). The functional dependence of the Gaussian on the x is through the quadratic form, $\Delta^2 = (x-\mu)^T\Sigma^{-1}(x-\mu)$, which

appears in the exponent in Eq. (B1). We consider a superposition of K Gaussian densities of the form, where the quantity $\Delta$ is called the *Mahalanobis distance* from $\mu$ to x and it reduces to the Euclidean distance when $\Sigma$ is the identity matrix (Bishop, 2006).

The joint distribution will be $p(z)p(x|z)$ and the marginal distribution of x is,

$$p(x) = \sum_z p(x,z) = \sum_z p(z)p(x|z) \tag{B2}$$

Here, $\sum_z p(x,z)$ is the probability distribution for the observations $x_1, ......., x_N$. So for every observed data point $x_n$, there is a corresponding latent variable $z_n$.

GMM represents the PDF as a weighted sum of K Gaussian classes as in Eq. (1). If we integrate Eq. (B1) with respect to x,

and note that both p(x) and Gaussian components are normalized we obtain,

$$\sum_{k=1}^{K} \lambda_k = 1 \tag{B3}$$

We call the parameters $\lambda_k$ mixing coefficients. The requirement $p(x) \geq 0$ together with $N(x|\mu_k,\Sigma_k) \geq 0$ implies $\lambda_k \geq 0$ for all k.

Combining these conditions, we can write, $0 \leq \lambda_k \leq 1$. The latent variable z is a K- dimensional binary random variable,

having a 1-of-K representation in which a particular element $z_k = 1$ and the rest are equal to 0. Therefore, $z_k \in \{0,1\}$ and $\sum_k z_k = 1$ and there are K possible states for the vector z according to which element is nonzero. The joint distribution $p(x,z)$ in terms of a marginal distribution $p(z)$ and a conditional distribution $p(x|z)$. The marginal distribution over z is specified in terms of the mixing coefficients $\lambda_k$, such that

$$p(z_k = 1) = \lambda_k$$

Because, z uses a 1-of-K representation, Eq. (1) can be written in the from

$$p(z) = \prod_{k=1}^{K} \lambda_k^{z_k} \tag{B4}$$

The conditional distribution of $x$ given a particular value for $z$ is a Gaussian

$$p(x|z_k = 1) = N(x|\mu_k, \Sigma_k) \tag{B5}$$

which can be written in the form,

$$p(x|z) = \prod_{k=1}^{K} N(x|\mu_k, \Sigma_k)^{z_k} \tag{B6}$$

The joint distribution will be $p(z)p(x|z)$ and the marginal distribution of x is,

$$
\begin{aligned}
p(x) = \sum_z p(x,z) &= \sum_z p(z)p(x|z) \\
&= \sum_{k=1}^{K} \lambda_k N(x|\mu_k, \Sigma_k) \\
&= \sum_{k=1}^{K} p(z_k = 1)p(x|z_k = 1) \\
&= \sum_{k=1}^{K} \lambda_k p_k(x)
\end{aligned}
\tag{B7}
$$

This equation is also called *Mixture distribution*.

Here, $p(x)$ stands for the observed PDF, and $\sum_z p(x,z)$ is the probability distribution for the observations $x_1, ......., x_N$. So
for every observed data point $x_n$, there is a corresponding latent variable $z_n$.

Gaussian mixture modelling nails down to an optimization problem that can be tackled by maximizing the likelihood of observed profiles. This optimization is referred to as a *model training*. It is solved with the Expectation- Maximization method. The conditional probability of z given x plays an important role in the Expectation-Maximization algorithm. $\gamma(z_k)$ represents $p(z_k = 1|x)$ whose value can be found using the Bayes theorem,

$$P(A|B) = \frac{P(B|A)P(A)}{P(B)}$$

So,

$$
\begin{aligned}
\gamma(z_k) \equiv p(z_k = 1|x) &= \frac{p(z_k = 1)p(x|z_k = 1)}{\sum_{k=1}^{K} p(z_k = 1)p(x|z_k = 1)} \\
&= \frac{\lambda_k N(x|\mu_k, \Sigma_k)}{\sum_{k=1}^{K} \lambda_k N(x|\mu_k, \Sigma_k)}
\end{aligned}
\tag{B8}
$$

Here, $\lambda_k$ is the prior probability of $z_k = 1$ and the quantity $\gamma(z_k)$ as the corresponding posterior probability once we have
observed x. The posterior probability for each component in GMM from which the data set was generated is called the *responsibilities*. Responsibilities sum to 1. This helps us predict which Gaussian is responsible for which data point.

Since the latent variables are never observed, and the correct values are not known in advance, Expectation Maximization is useful to figure out what z represents without someone to specify it beforehand.

EM method aims to iteratively improve the results based on some initial assumptions on the mean, standard deviation, and
latent values. Every single iteration is of two steps - the so E step and the M step.

In the *expectation* step, it uses current values for the parameters to evaluate the posterior probabilities or responsibilities given by Eq. (B8). We then use these probabilities in the *maximization* step to re-estimate the means, covariances, and mixing coefficients.

**EM for Gaussian Mixture**

1. Initialization of the parameters and evaluate the initial values for log-likelihood. Parameters are: Means $\mu_k$, covariances $\Sigma_k$ and mixing coefficients $\lambda_k$

2. **E step :** Evaluation of the responsibilities using the current parameter values.

$$\gamma(z_{ik}) == \frac{\lambda_k N(x_i|\mu_k, \Sigma_k)}{\sum_{k=1}^{K} \lambda_k N(x_i|\mu_k, \Sigma_k)}$$

3. **M step:** Re-estimate the parameters using the current responsibilities

$\cdot$ $\mu_k^{new} = \frac{1}{N_k} \sum_{i=1}^{N} \gamma(z_{ik})x_i$

$\cdot$ $\Sigma_k^{new} = \frac{1}{N_k} \sum_{i=1}^{N} \gamma(z_{ik})(x_i - \mu_k^{new})(x_i - \mu_k^{new})^T$

$\cdot$ $\lambda_k^{new} = \frac{N_k}{N}$

where, $N_k = \sum_{k}^{N} \gamma(z_{ik})$

4. Evaluate the log likelihood

$$\ln p(X|\lambda, \mu, \Sigma) = \sum_{i=1}^{N} \ln \{ \sum_{k=1}^{K} \lambda_k N(x_i|\mu_k, \Sigma_k) \}$$

and check for convergence of either the parameters or the log-likelihood. If the convergence criterion is not satisfied, return to step 2.

**Appendix C: Selecting the number of classes**

The main free input parameter to the model training procedure is the number of mixture components $K$. Determining
the most appropriate number of components automatically is a difficult problem that often contains a degree of subjectivity,

requiring domain expertise. Here we use a combination of statistical guidance and expert judgment to select the number of classes.

For statistical guidance, we use BIC, which stands for the *Bayesian Information Criterion*. The BIC is an empirical approach to the model probability computed as:

$$BIC(K) = -2\ell(K) + N_f(K)log(n) \tag{C1}$$

where $\ell(K)$ is the log-likelihood of the trained model with $K$ classes, $n$ is the number of profiles used in the BIC test. The log-likelihood function as below,

$$\ell = \ln p(X|\lambda, \mu, \Sigma) = \sum_{i=1}^{N} \ln \left\{ \sum_{k=1}^{K} \lambda_k N(x_i|\mu_k, \Sigma_k) \right\}$$
$$= \sum_{i=1}^{N} \ln \sum_{k=1}^{K} (\lambda_k p(x_i)) \tag{C2}$$

The log-likelihood of the data set, assuming independent observations, is:

$$\ell(\theta) = \sum_{i=1}^{N} log p(x_i; \theta), \tag{C3}$$

where it is explicit that the log-likelihood is a function of the set of parameters $\theta$, and where $p(x_i; \theta)$ is the probability given in equation C2 for the data set instance $x_i$ using the parameters $\theta$. $N_f$ is the number of the independent parameters to be estimated (the sum of the component weights, Gaussian means, and covariance matrix elements in the $d$-dimensional data space after PCA our new dimension is d):

$$N_f(k) = (K-1) + Kd + \frac{Kd(d-1)}{2} \tag{C4}$$

The BIC is empirical; the first r.h.s term in equation C1 decreases as the likelihood of the statistical model increases, while the second r.h.s term is a penalty term that increases with $K$ and thus discourages over-fitting (Maze et al., 2017). The "ideal" value for $K$, in terms of this statistical metric, would be one that minimizes BIC, i.e., the likelihood of the model has been maximized without overfitting. One may also find that the BIC curve "plateaus", indicating that the model has reached maximum likelihood, i.e., further increases in the statistical complexity of the model no longer noticeably improve the likelihood. Empirical approaches like BIC are often used in statistics, especially when constraining the parameters is difficult or subjective. They can give us a rough estimate of what data collection might look like if we were able to survey the entire population (Maze et al., 2017).

Here, $\theta = \{\lambda, \mu, \Sigma\}$ is the set of parameters that minimize the misfit between the PDF of the data set that is going to be used for calculation and the PDF of the original data set. To train a GMM, i.e., to maximize $\ell(\theta)$ with regard to $\theta$ so that our BIC can be lowest, we need a data set $x$ and a given number of components $K$ (Maze et al., 2017).

## Appendix D: Maximum ozone concentration

Here we provide detailed information about the maximum ozone concentration based on seasons.

| Class | Hist (lev) [mPa] | (mean) | (std) | SSP1-2.6 (lev) [mPa] | (mean) | (std) | SSP5-8.5 (lev) [mPa] | (mean) | (std) |
|---|---|---|---|---|---|---|---|---|---|
| 1 | 50 | 13.600 | 1.200 | 50 | 15.600 | 1.200 | 100 | 14.600 | 1.100 |
| 2 | 30 | 13.800 | 0.600 | 50 | 16 | 3.900 | 30 | 14 | 1 |
| 3 | 20 | 14.400 | 0.900 | 20 | 14.700 | 0.600 | 20 | 13.800 | 0.700 |
| 4 | 30 | 13.700 | 1.700 | 30 | 14.300 | 2.600 | 20 | 13.800 | 3.100 |
| 5 | 50 | 17.800 | 4 | 50 | 20 | 4.500 | 70 | 21.100 | 4 |
| 6 | 70 | 17.700 | 1.100 | 70 | 20.700 | 1.300 | 70 | 22 | 3.400 |

**Table D1.** Pressure level (lev) of the maximum value of class mean ozone concentration during DJF. The mean and standard deviation values of the class statistics are given in mPa.

| Class | Hist (lev) [mPa] | (mean) | (std) | SSP1-2.6 (lev) [mPa] | (mean) | (std) | SSP5-8.5 (lev) [mPa] | (mean) | (std) |
|---|---|---|---|---|---|---|---|---|---|
| 1 | 50 | 15.300 | 0.600 | 70 | 16.800 | 0.700 | 50 | 16.500 | 0.600 |
| 2 | 50 | 14 | 2 | 50 | 15.100 | 2.400 | 50 | 15.400 | 2.300 |
| 3 | 20 | 15.100 | 0.800 | 20 | 15.400 | 0.600 | 20 | 14.300 | 1 |
| 4 | 30 | 14.100 | 2.100 | 30 | 14.600 | 3 | 30 | 14.400 | 3.600 |
| 5 | 50 | 16.900 | 2 | 70 | 21.500 | 3.900 | 70 | 23.900 | 3.200 |
| 6 | 50 | 16.200 | 1.900 | 70 | 20.800 | 4 | 70 | 21 | 4.900 |

**Table D2.** same as Table D1, except for MAM

| Class | Hist (lev) [mPa] | (mean) | (std) | SSP1-2.6 (lev) [mPa] | (mean) | (std) | SSP5-8.5 (lev) [mPa] | (mean) | (std) |
|---|---|---|---|---|---|---|---|---|---|
| 1 | 70 | 17 | 1.100 | 70 | 18.800 | 1.400 | 70 | 19.300 | 1.100 |
| 2 | 50 | 15.800 | 3 | 50 | 18.800 | 3.800 | 50 | 17.900 | 5.200 |
| 3 | 20 | 15.400 | 1 | 20 | 15.500 | 0.800 | 20 | 14.900 | 1.300 |
| 4 | 30 | 14.300 | 2.100 | 30 | 14.800 | 2.600 | 20 | 14.300 | 2.900 |
| 5 | 0 | 0 | 0 | 30 | 16.400 | 2.100 | 50 | 22.600 | 1.900 |
| 6 | 50 | 14 | 1.500 | 70 | 15.700 | 2.300 | 50 | 15.400 | 2.500 |

**Table D3.** same as Table D1, except for JJA.

| Class | Hist (lev) [mPa] | (mean) | (std) | SSP1-2.6 (lev) [mPa] | (mean) | (std) | SSP5-8.5 (lev) [mPa] | (mean) | (std) |
|---|---|---|---|---|---|---|---|---|---|
| 1 | 30 | 9.600 | 3.300 | 70 | 18.300 | 3.200 | 70 | 21.200 | 3.400 |
| 2 | 50 | 16 | 1.700 | 70 | 16.600 | 2.500 | 30 | 16.600 | 3.900 |
| 3 | 20 | 15.300 | 1 | 20 | 15.500 | 0.800 | 20 | 15.300 | 1.100 |
| 4 | 30 | 14.300 | 1.600 | 30 | 14.900 | 2.400 | 30 | 14.300 | 1.800 |
| 5 | 0 | 0 | 0 | 0 | 0 | 0 | 0 | 0 | 0 |
| 6 | 50 | 13.900 | 2 | 50 | 14.700 | 2.500 | 50 | 15.500 | 2.500 |

**Table D4.** same as Table D1, except for SON.

*Code and data availability.* Data from UKESM1 is part of the CMIP6 data suite, which is freely available from a number of sources. For this study, we used Pangeo (https://pangeo.io/) for rapid data access and averaging. We used "preprocessing" script from Julian Busecke (https://github.com/jbusecke/cmip6_preprocessing). All scripts used to data process and produce figures for this paper are online via Zenodo (Fahrin and Jones, 2023)

.

*Author contributions.* DCJ designed the initial project and developed much of the software. FF performed the analysis, worked with the software, and created the figures. JK and ATA provided expert guidance on analyzing the results and placing them in the wider context of atmospheric chemistry. FF and DCJ wrote the initial manuscript, JK edited the introduction, and all authors assisted with edits.

*Competing interests.* The contact author and the co-authors do not have any competing interest.

*Acknowledgements.* This work originated as a master's project in the Department of Mathematical Sciences at Georgia Southern University. The authors wish to thank Guillaume Maze for suggesting the particular training dataset method used here. We acknowledge the use of the Pangeo platform in obtaining our data (https://pangeo.io/). DJ acknowledges funding from a UKRI Future Leaders Fellowship (reference MR/T020822/1) and the North Atlantic Climate System Integrated Study (ACSIS) (grant NE/N018028/1). JK and ATA thank the Met Office CSSP-China programme for funding the POzSUM project. The authors thank two anonymous reviewers for their helpful comments on the manuscript.

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
