# Peer review of "Technical Note: Unsupervised classification of ozone profiles in UKESM1"

_Atmospheric Chemistry and Physics, 2022_

## Author Comment (AC1)

General comments:

1. Since the chemical and dynamical processes controlling ozone concentrations in the stratosphere are quite different from those controlling near-surface ozone, what is the rationale for performing the GMM classification on the entire ozone profile? Could you instead cluster different vertical regions, such as stratosphere or troposphere, separately? It seems like the results might be easier to interpret and the clusters more applicable to model comparisons of specific features like surface concentration if signals from near-surface processes weren't mixed together with signals from stratospheric circulation in the creation of the clusters.

Thank you for this suggestion. In our revised approach, we have omitted the near-surface values before carrying out the classification step, which has indeed given us a set of classes that are somewhat easier to interpret. The highest pressure in our analysis is now 850 hPa. Our approach does still combine some tropospheric values with some stratospheric values, but the influence of the near-surface has been greatly reduced. As an extra advantage, our new classification approach is able to cover a larger fraction of the global model domain, as we have not had to discard profiles with surface pressures less than 1000 dbar.

Although it could be insightful to carry out completely separate stratospheric and tropospheric clustering analysis, we are interested in characterizing the entire ozone profile in the mid-to-upper troposphere and the entire stratosphere, because such an approach makes use of more of the data and will therefore be more general. Analyzing the entire profile also gets around the problem of having to decide exactly where to separate the troposphere from the stratosphere; given that the tropopause is gentler in some regions than others, the boundary between the two is not always easy to unambiguously determine. In addition, there are not very many pressure levels in the stratosphere, which would complicate any attempts to cluster there. We hope that you find our solution of discarding the near-surface pressure levels to be acceptable.

2. What is the advantage of using the GMM clustering method over just grouping profiles by e.g. tropopause height or altitude of peak ozone, since these seem to be prominent features distinguishing the derived classifications? It is encouraging to see that the GMM analysis leads to results that are consistent with known sources of variability, but to justify the complexity of this GMM approach, it would be helpful to also highlight specific cases where the GMM creates a more meaningful classification than could be obtained with a single variable such as tropopause height.

GMM takes the entire structure, within the selected pressure range, of all ozone profiles in the training dataset into account, which makes it more general than sorting or grouping the profiles based on a single value (e.g. tropopause height). Single-value classification schemes would have to rely on ad-hoc decisions about "cutoff" values between one group and another (e.g. low tropopause vs high tropopause). Although statistical distributions may offer some guidance there, generally speaking, we can't expect such ad-hoc, specific cutoff values to be applicable across many different numerical model experiments, especially when considering different future climate scenarios, over which the cutoff values separating one class from another may shift. In contrast, by taking the entire structure of ozone profiles into account, GMM offers an approach that is insensitive to specific choices about ad-hoc cutoff values. In addition, the fact that, in our revised analysis, the classes vary with season makes them more useful and informative than a simple latitude-based classification scheme.

As you have said, the fact that GMM returns an intuitive, reasonably interpretable set of classes across several different numerical experiments indicates that it is generally applicable; we don't have to select cutoff values. In addition, although we have not made much use of this fact in this manuscript, the GMM approach is probabilistic; it returns a set of probabilities across classes that could be used to examine boundaries between classes in a probabilistic fashion, which has been done in the Southern Ocean (e.g. Thomas et al., 2021, https://doi.org/10.5194/os-17-1545-2021). This could be an avenue for future exploration.

Specific Comments:

1.Lines 46-49: Please provide a reference.

Thank you for the suggestion.

We have added several references here:

However, an acceleration of the Brewer-Dobson circulation (BDC) associated with increasing greenhouse gas concentrations may lead to reductions in lower tropical stratospheric ozone mixing ratios (Eyring et al., 2013; Meul et al., 2016; Keeble et al., 2017) while increasing transport of ozone into the mid-latitudes troposphere (Banerjee et al., 2016; Meul et al., 2018).

2. The discussion of previous work on ozone clustering could be expanded.

We have expanded the discussion. Now the discussion reads:

Clustering techniques have already been used in ozone concentration studies for understanding long-term variability. Boleti et al. (2020) have applied a multidimensional clustering technique to understand the long-term trend of ozone. Diab et al. (2004) used a six clusters analysis which resulted in distinct clusters of "background" and "polluted" with below and above ozone mixing ratios from over 100 ozonesonde profiles launched from a subtropical Southern Hemisphere Additional Ozonesondes (SHADOZ) (Thompson et al., 2003) site, Irene, South Africa. Jensen et al. (2012) performed a cluster analysis named self-organizing maps (SOM) (Kohonen, 2012) on over 900 tropical ozonesonde profiles. Their findings with four-cluster results were similar to Diab et al. (2004). Both studies showed that the seasonal influences of biomass burning and convection dominate ozone variability. Stauffer et al. (2016) documented the influence of meteorological conditions on the shape of the ozone profile from the troposphere to the lower stratosphere by applying the SOM clustering technique to ozonesonde data from specific northern hemisphere midlatitude geographical regions. Later they expanded the study for global ozonesonde sites to show the variation of ozone profiles cluster for various regions and how they vary based on meteorology and chemistry depending on latitudes (Stauffer et al., 2018).

3. Lines 97-98: The requirement of surface pressure reaching 1000 hPa seems like a significant limitation. Would the results be much different (and the coverage increase) if you used something like 900 hPa instead?

Yes, thank you for this excellent suggestion. We have discarded pressure levels above 850 hpa from our data set. Now the results are based on pressure levels in the range 1-850 hpa pressure level. The resulting clusters cover more area overall and are easier to interpret.

4. Line 130: How does the pressure level standardization affect the relative importance of the stratospheric versus the tropospheric portions of the profile in determining the clusters?

Ozone concentration measured from different pressure levels does not contribute equally to the analysis because of large differences in tropospheric and stratospheric ozone values, and this might end up creating a bias in the algorithm. The primary purpose of standardization is to put ozone concentrations from different pressure levels on the same scale.

In standardizing ozone on each pressure level separately, we are allowing variations in the stratosphere to have the same impact as variations in the troposphere relative to the usual variability found on each pressure level. If we didn't do this, then our classes would simply be determined by the pressure levels on which the variability is highest, in absolute terms.

5. Line 149: Define BIC and refer the reader to the description in the appendix.

We changed the statement.

6. Lines 194-201: Do the higher tropopause and higher surface values both contribute to the definition of this cluster, or is it just that the clusters vary strongly with latitude (as shown in Fig. 4) and many other features also co-vary with latitude?

It is important to note that the algorithm does not have any information about the latitude of the profiles. Our implementation of GMM only uses the ozone values themselves. The classification will indeed be influenced by the entire structure of the ozone profile, especially since the ozone profiles have been standardized on each pressure level. In any case, now that we have excluded the near-surface values, the near-surface no longer has an influence on the classification.

7. Line 216: is mPa the right unit here?

We used mPa everywhere in this study to keep consistency.

8.Line 219: Replace "reasonable" with something more quantitative

Changed to "known". This isn't more quantitative, but we hope that you find it suitable.

9. Fig 4 (and 5) and Fig 4 caption: Does "median" make sense with respect to classifications here? Are the classes quantitatively ordered such that class 3 is in between classes 2 and 4? Also, is there much temporal variability (within the decade) in what class a particular grid box falls in? If so, it would be nice to show that since it could help clarify how the GMM classification differs from a purely latitude-based classification.

The label maps have been changed; we now examine seasonal variability.

10. Line 249: Does this mean the fact that class 1 has the lowest ozone or the fact that the class 1 ozone is lower in the historic run is consistent with the reduction with precursors?

It is consistent with the reduction of precursors. We have changed the statement to:

As with the historical experiment, class 1 has the lowest surface ozone (Table 1), which is consistent with the reduction in surface ozone precursors in this experiment. The maximum value of stratospheric ozone increases under this scenario, which is a signature of the recovery of the ozone hole (Keeble et al., 2021).

We hope you find it suitable.

11. Lines 272-275: Is this explanation proven by your analysis or just consistent with your results?

It is consistent with our results. We used references from studies that have shown these.

12. Lines 284-285: Please explain how this conclusion is reached from Figs 4-5

We have changed the statement since we are now doing the seasonal averages.

13. Lines 295-297: Is it possible to relate this quantitatively to the extent of the model's Hadley cell?

Possibly, although an in-depth analysis of the circulation of the model is beyond the scope of this technical note. We only aim to highlight and demonstrate this ozone classification approach.

14. Lines 298-304: Are these results different from what would be inferred with latitudinal averages?

   These results should be consistent with latitudinal averages. What is different here is we did not use the latitudinal information for our algorithm, still our result is consistent with what we expect from latitudinal averages.

15. Line 321: This statement needs more support. Relate to Table 4?

We agree that this statement was vague, and we have removed it from the paper.

We have made the technical changes to the manuscript. We wish to thank the reviewers for helpful insights. Your comment helped us to improve our manuscript.

---

## Author Comment (AC2)

**Major comments:**

1. This analysis uses annual mean ozone profiles, which provide no information on ozone's high day-to-day variability in the troposphere. The method is also applied to full profiles that include the troposphere and stratosphere. The result is a very smooth ozone field dominated by the stratosphere. As I describe below, the resulting clusters seem to be insensitive to prominent tropospheric ozone features. For this reason, I think the analysis needs to be applied to the troposphere and the stratosphere separately.

Yes, thank you for these suggestions. For our application, we are not specifically interested in day-to-day variability in the atmosphere, but we agree that including more temporal information in our classification approach would be prudent. We have modified our approach to include seasonal variations by switching from annual mean profiles to seasonal mean profiles. Furthermore, we have also excluded some of the near-surface pressure levels, such that the classification is less affected by near-surface ozone processes. The suggestion to classify the troposphere and stratosphere separately is interesting; unfortunately, there are not enough pressure levels in the stratosphere alone to justify classifying it entirely separately from the troposphere. In any case, we chose this approach because it retains more of the entire profile structure, which we view as an advantage - we want the resulting classes to be influenced by the upper troposphere, the stratosphere, and the interaction between the two.

2. What is the impact (or limitation) of using annual averages? Ozone concentrations vary widely from summer to winter in both the troposphere and the stratosphere. How different are the clusters if the analysis is applied separately to summer and winter months? Another problem with the annual average is that mid-latitudes are heavily influenced by polar air masses in winter (low tropopause), and by tropical air masses in summer (high tropopause). So the annual average is just an unrealistic homogenization of very different air masses, and does not reflect the typical ozone profiles one might find in any given month or season.

Yes, this is an important point. We have switched to classifying seasonal mean profiles, which allows seasonal variation. The analysis in the paper has been updated to reflect this seasonal variation. Thank you for the suggestion.

The analysis makes no use of observations, and with no evaluation against real-world data we are unable to understand the accuracy of the method. Stauffer et al. (2018) clustered ozone profiles at more than 2 dozen ozonesonde stations worldwide. I realize the authors can't use sparse observations as the basis for this global-scale analysis, but they can certainly evaluate the results against observations. The authors should examine the observed profiles above the ozonesonde stations that lie within each of the clustered regions. Do the profiles within each region have similar characteristics? If so, then the method is applicable to the real-world; if not, then the usefulness of the method is questionable. What is the result when the observations are then examined by season? Are the observations within each cluster similar to each other in summer, and also in winter? Or does everything break down (see my comment above about seasonal variability in the mid-latitudes).

In this paper, we are focusing on GMM as a model analysis and comparison tool. We agree that applications to observational data would be interesting, but it is beyond the scope of this short technical note. In any case, it is not necessarily obvious how one would carry out such a comparison, given the sparse observational coverage of atmospheric ozone. Specifically, one might try to fit the GMM using observational profiles only and then use those classes to validate and analyze a model run. This has been done for ocean temperature profiles for a specific region in the European Arctic (Thomas and Müller, 2022, https://doi.org/10.1016/j.ocemod.2022.102092). However, given that observational coverage of atmospheric ozone is not especially uniform, it would be difficult to generate a sufficiently general training dataset for GMM. Any resulting GMM would be biased towards the ozone profiles seen at the observing site. We believe we have shown that GMM can, at the very least, be a useful tool for model ozone analysis and comparison.

Other comments:

1. Line 59: When reviewing clustering techniques as applied to ozone profiles, the authors should include Stauffer et al. (2016, 2018).

Thank you for providing some excellent references. We cited them and extended our ozone clustering techniques applied in other studies. The paragraph now reads:

Clustering techniques have already been used in ozone concentration studies for understanding long-term variability. Boleti et al. (2020) have applied a multidimensional clustering technique to understand the long-term trend of ozone. Diab et al. (2004) used a six-cluster analysis which resulted in distinct clusters of "background" and "polluted" with below and above ozone mixing ratios from over 100 ozonesonde profiles launched from a subtropical Southern Hemisphere Additional Ozonesondes (SHADOZ) (Thompson et al., 2003) site, Irene, South Africa. Jensen et al. (2012) performed a cluster analysis named self-organizing maps (SOM) (Kohonen, 2012) on over 900 tropical ozonesonde profiles. Their findings with four-cluster results were similar to Diab et al. (2004). Both studies showed that the seasonal influences of biomass burning and convection dominate ozone variability. Stauffer et al. (2016) documented the influence of meteorological conditions on the shape of the ozone profile from the troposphere to the lower stratosphere by applying SOM clustering technique to ozonesonde data from specific northern hemisphere midlatitude geographical regions. Later they expanded the study for global ozonesonde sites to show the variation of ozone profiles cluster for various regions and how they vary based on meteorology and chemistry depending on latitudes (Stauffer et al., 2018).

2. Line 61: The perceived methodology and aim of Chang et al. 2017, as stated in the manuscript, is not correct. Chang et al. 2017 are not seeking to cluster similar ozone monitoring sites. Rather they are trying to quantify the regional-scale, long-term trend of ozone while accounting for the spatial distribution of the sites and the correlation between sites. This method accounts for the uneven distribution of sites and prevents any heavily-sampled sub-region from exerting an out-sized influence on the trend

We agree that the citation was not correct. Thank you for pointing that out. We got rid of that part.

3. Line 95-98: I don't understand why the study is limited to 1-1000 hPa. This omits a large section of the globe, i.e., land regions more than 100-200 m above sea level. I realize the method cannot tolerate missing values, but why not conduct the study for profiles in the range of 1-950 hPa; this way, you retain most of the land areas.

 Thank you for the suggestion. We are now focusing on 1-850 hPa to retain more land areas.

4. Line 110: This statement is problematic:"The motivation behind withholding the geographical information is that there is no reason for the vertical ozone structure of the profile to be unique to a given region (Maze et al., 2017)." Using a paper that deals with ocean temperature, the authors

seem to suggest that there is no discernable structure in the global ozone distribution and that one region is no different from another. Yet, plenty of observation-based studies identify clear structure in the global ozone distribution that varies with season [Kley et al., 1996; Thouret et al., 1998; Oltmans et al., 1996, 2004; Thompson et al., 2003; Cooper et al., 2007; Gaudel et al., 2020;]. Therefore, certain profile types are more likely to occur in some regions than in other regions. This statement needs to be revised.

We agree that the original statement was unclear. We have revised the text and hope that you find the new statement suitable.

5. Line 173: To say that the tropopause is around 300 hPa is a gross over-simplification. As can be seen in Figure 2, there are plenty of profiles in which the tropopause is around 150 hPa, which is common in the tropics.

We have changed this statement to :

The tropopause height, above which the ozone starts increasing, varies between 300-150 hpa depending on the location of the profiles.

6. Line 175: The statement that ozone increases near the surface is problematic because ozone is plotted in units of mPa. If ozone is plotted in units of ppbv (the typical unit for evaluating air pollution levels in the troposphere), then we would see that the average ozone profile has more ozone in the upper troposphere, especially at mid-and high latitudes (see the ozone profile papers that I cited above). Furthermore, Jaffe and Wigder (2012) is not a sufficient reference because they only discuss ozone at the surface and do not mention the vertical distribution of ozone.

We use ozone partial pressures (mPa) in this study because if we use mixing ratios the ozone profiles would span many orders of magnitude, with surface ozone mixing ratios in the 10s of ppb, and stratospheric ozone mixing ratios reaching a maximum of ppm, and so the profiles would be dominated by the shape in the stratosphere, and it would be very difficult to see what is happening in the troposphere. The choice of mPa shows more clearly the relative structure between profiles in the troposphere. We have amended this paragraph to explicitly highlight the fact that we are discussing ozone concentrations and replaced the reference with that of Monks et al., which provides a more complete review of tropospheric ozone distributions and processes. The paragraph now reads:

Our purpose is to identify coherent patterns within the collection of profiles using unsupervised machine learning. Overall, the profiles reveal relatively high ozone concentrations in the lower and middle stratosphere which peak and then decrease gradually in the upper stratosphere. The tropopause height, above which the ozone concentrations start increasing, varies between 300-150 hPa depending on the location of the profiles. The peak starts decreasing at around 70 hPa and higher altitudes above (Figure 2). In the troposphere, ozone concentrations are fairly constant and then increases towards the surface, in part due the availability of ozone precursors from biomass burning and anthropogenic emissions sources (e.g., Monks et al., 2015).

7. Line 216: Why is the high surface ozone only attributed to biomass burning? This cluster spans the major fossil fuel combustion regions of the northern hemisphere, which are known to drive ozone production across the region.

This has been amended and now refers to ozone precursors from different sources. The sentence now reads:

In the troposphere, ozone concentrations are fairly constant and then increase towards the surface, in part due to the availability of ozone precursors from biomass burning and anthropogenic emissions sources (e.g., Monks et al., 2015).

8. Line 265: The statement that ozone precursor emissions generally increase under SSP5-8.5 isn't really correct, as emissions continue to decrease in developed nations but increase in the developing world. This discussion should also consider the findings of Zanis et al., 2022.

This statement has been expanded upon in the final draft, and we have also included some of the discussion from Zanis et al.. The relevant section now reads:

Here we examine the structure of atmospheric ozone in the 2095-2100 years of the SSP5-8.5 experiment. In this experiment ozone mixing ratios are generally higher throughout much of the troposphere and upper stratosphere. In the troposphere, the drivers of this increase are complex. Under the assumptions of the SSP5-8.5 scenario, global mean emissions of NOx and CO are lower in 2095 than the present day, while global mean emissions of CH4 are higher (Gidden et al., 2019). However, changes to ozone precursor emissions alone do not drive tropospheric ozone changes, which is also affected fby climate change, with increasing tropospheric temperatures changing biogenic volatile organic compounds (BVOC) emissions, the availability of tropospheric water vapor, and stratosphere-to-troposphere transport of ozone, which taken together drive increases to tropospheric ozone concentrations (e.g., Griffiths et al., 2021; Turnock et al., 2021; Zanis et al., 2022). In the stratosphere this increase is simpler to understand. Upper stratospheric ozone

increases under all SSPs as ozone depleting substances decrease, but increases more in scenarios which assume larger increases in greenhouse gas emissions due to the resulting CO2-induced cooling of the stratosphere and the impacts this has on gas phase chemistry (e.g., Haigh and Pyle, 1982; Jonsson et al., 2004).

---

## Author Response (AR3)

The authors wish to thank the reviewers for their careful consideration of this manuscript and their helpful feedback. We have revised the manuscript and have attempted to address all reviewer comments below.

**Major Revision:**

**Response to RC1:**

General comments:

1. Since the chemical and dynamical processes controlling ozone concentrations in the stratosphere are quite different from those controlling near-surface ozone, what is the rationale for performing the GMM classification on the entire ozone profile? Could you instead cluster different vertical regions, such as stratosphere or troposphere, separately? It seems like the results might be easier to interpret and the clusters more applicable to model comparisons of specific features like surface concentration if signals from near-surface processes weren't mixed together with signals from stratospheric circulation in the creation of the clusters.

Thank you for this suggestion. In our revised approach, we have omitted the near-surface values before carrying out the classification step, which has indeed given us a set of classes that are somewhat easier to interpret. The highest pressure in our analysis is now 850 hPa. Our approach does still combine some tropospheric values with some stratospheric values, but the influence of the near-surface has been greatly reduced. As an extra advantage, our new classification approach is able to cover a larger fraction of the global model domain, as we have not had to discard profiles with surface pressures less than 1000 dbar.

Although it could be insightful to carry out completely separate stratospheric and tropospheric clustering analysis, we are interested in characterizing the entire ozone profile in the mid-to-upper troposphere and the entire stratosphere, because such an approach makes use of more of the data and will therefore be more general. Analyzing the entire profile also gets around the problem of having to decide exactly where to separate the troposphere from the stratosphere; given that the tropopause is gentler in some regions than others, the boundary between the two is not always easy to unambiguously determine. In addition, there are not very many pressure levels in the stratosphere, which would complicate any attempts to cluster there. We hope that you find our solution     of     discarding     the     near-surface     pressure     levels     to     be     acceptable.

2. What is the advantage of using the GMM clustering method over just grouping profiles by e.g. tropopause height or altitude of peak ozone, since these seem to be prominent features distinguishing the derived classifications? It is encouraging to see that the GMM analysis leads to results that are consistent with known sources of variability, but to justify the complexity of this GMM approach, it would be helpful to also highlight specific cases where the GMM creates a more meaningful classification than could be obtained with a single variable such as tropopause height.

GMM takes the entire structure, within the selected pressure range, of all ozone profiles in the training dataset into account, which makes it more general than sorting or grouping the profiles based on a single value (e.g. tropopause height). Single-value classification schemes would have to rely on ad-hoc decisions about "cutoff" values between one group and another (e.g. low tropopause vs high tropopause). Although statistical distributions may offer some guidance there, generally speaking, we can't expect such ad-hoc, specific cutoff values to be applicable across many different numerical model experiments, especially when considering different future climate scenarios, over which the cutoff values separating one class from another may shift. In contrast, by taking the entire structure of ozone profiles into account, GMM offers an approach that is insensitive to specific choices about ad-hoc cutoff values. In addition, the fact that, in our revised analysis, the classes vary with season makes them more useful and informative than a simple latitude-based classification scheme.

As you have said, the fact that GMM returns an intuitive, reasonably interpretable set of classes across several different numerical experiments indicates that it is generally applicable; we don't have to select cutoff values. In addition, although we have not made much use of this fact in this manuscript, the GMM approach is probabilistic; it returns a set of probabilities across classes that could be used to examine boundaries between classes in a probabilistic fashion, which has been done in the Southern Ocean (e.g. Thomas et al., 2021, https://doi.org/10.5194/os-17-1545-2021). This could be an avenue for future exploration.

Specific Comments:

1.Lines 46-49: Please provide a reference.

Thank you for the suggestion.

We have added several references here:

However, an acceleration of the Brewer-Dobson circulation (BDC) associated with increasing greenhouse gas concentrations may lead to reductions in lower tropical stratospheric ozone mixing ratios (Eyring et al., 2013; Meul et al., 2016; Keeble et al., 2017) while increasing transport of ozone into the mid-latitudes troposphere (Banerjee et al., 2016; Meul et al., 2018).

2. The discussion of previous work on ozone clustering could be expanded.

We have expanded the discussion. Now the discussion reads:

Clustering techniques have already been used in ozone concentration studies for understanding long-term variability. Boleti et al. (2020) have applied a multidimensional clustering technique to understand the long-term trend of ozone. Diab et al. (2004) used a six clusters analysis which resulted in distinct clusters of "background" and "polluted" with below and above ozone mixing ratios from over 100 ozonesonde profiles launched from a subtropical Southern Hemisphere Additional Ozonesondes (SHADOZ) (Thompson et al., 2003) site, Irene, South Africa. Jensen et al. (2012) performed a cluster analysis named self-organizing maps (SOM) (Kohonen, 2012) on over 900 tropical ozonesonde profiles. Their findings with four-cluster results were similar to Diab et al. (2004). Both studies showed that the seasonal influences of biomass burning and convection dominate ozone variability. Stauffer et al. (2016) documented the influence of meteorological conditions on the shape of the ozone profile from the troposphere to the lower stratosphere by applying the SOM clustering technique to ozonesonde data from specific northern hemisphere midlatitude geographical regions. Later they expanded the study for global ozonesonde sites to show the variation of ozone profiles cluster for various regions and how they vary based on meteorology and chemistry depending on latitudes (Stauffer et al., 2018).

3. Lines 97-98: The requirement of surface pressure reaching 1000 hPa seems like a significant limitation. Would the results be much different (and the coverage increase) if you used something like 900 hPa instead?

Yes, thank you for this excellent suggestion. We have discarded pressure levels above 850 hpa from our data set. Now the results are based on pressure levels in the range 1-850 hpa pressure level. The resulting clusters cover more area overall and are easier to interpret.

4. Line 130: How does the pressure level standardization affect the relative importance of the stratospheric versus the tropospheric portions of the profile in determining the clusters?

Ozone concentration measured from different pressure levels does not contribute equally to the analysis because of large differences in tropospheric and stratospheric ozone values, and this might end up creating a bias in the algorithm. The primary purpose of standardization is to put ozone concentrations from different pressure levels on the same scale.

In standardizing ozone on each pressure level separately, we are allowing variations in the stratosphere to have the same impact as variations in the troposphere relative to the usual variability found on each pressure level. If we didn't do this, then our classes would simply be determined by the pressure levels on which the variability is highest, in absolute terms.

5. Line 149: Define BIC and refer the reader to the description in the appendix.

We changed the statement.

6. Lines 194-201: Do the higher tropopause and higher surface values both contribute to the definition of this cluster, or is it just that the clusters vary strongly with latitude (as shown in Fig. 4) and many other features also co-vary with latitude?

It is important to note that the algorithm does not have any information about the latitude of the profiles. Our implementation of GMM only uses the ozone values themselves. The classification will indeed be influenced by the entire structure of the ozone profile, especially since the ozone profiles have been standardized on each pressure level. In any case, now that we have excluded the near-surface values, the near-surface no longer has an influence on the classification.

7. Line 216: is mPa the right unit here?

We used mPa everywhere in this study to keep consistency.

8.Line 219: Replace "reasonable" with something more quantitative

Changed to "known". This isn't more quantitative, but we hope that you find it suitable.

9. Fig 4 (and 5) and Fig 4 caption: Does "median" make sense with respect to classifications here? Are the classes quantitatively ordered such that class 3 is in between classes 2 and 4? Also, is there much temporal variability (within the decade) in what class a particular grid box falls in? If so, it would be nice to show that since it could help clarify how the GMM classification differs from a purely latitude-based classification.

The label maps have been changed; we now examine seasonal variability.

10. Line 249: Does this mean the fact that class 1 has the lowest ozone or the fact that the class 1 ozone is lower in the historic run is consistent with the reduction with precursors?

It is consistent with the reduction of precursors. We have changed the statement to:

As with the historical experiment, class 1 has the lowest surface ozone (Table 1), which is consistent with the reduction in surface ozone precursors in this experiment. The maximum value of stratospheric ozone increases under this scenario, which is a signature of the recovery of the ozone hole (Keeble et al., 2021).

We hope you find it suitable.

11. Lines 272-275: Is this explanation proven by your analysis or just consistent with your results?

It is consistent with our results. We used references from studies that have shown these.

12. Lines 284-285: Please explain how this conclusion is reached from Figs 4-5

We have changed the statement since we are now doing the seasonal averages.

13. Lines 295-297: Is it possible to relate this quantitatively to the extent of the model's Hadley cell?

Possibly, although an in-depth analysis of the circulation of the model is beyond the scope of this technical note. We only aim to highlight and demonstrate this ozone classification approach.

14. Lines 298-304: Are these results different from what would be inferred with latitudinal averages?

   These results should be consistent with latitudinal averages. What is different here is we did not use the latitudinal information for our algorithm, still our result is consistent with what we expect from latitudinal averages.

15. Line 321: This statement needs more support. Relate to Table 4?

We agree that this statement was vague, and we have removed it from the paper.

We have also made the technical changes to the manuscript.

**Response to RC2:**

**Major comments:**

1. This analysis uses annual mean ozone profiles, which provide no information on ozone's high day-to-day variability in the troposphere. The method is also applied to full profiles that include the troposphere and stratosphere. The result is a very smooth ozone field dominated by the stratosphere. As I describe below, the resulting clusters seem to be insensitive to prominent tropospheric ozone features. For this reason, I think the analysis needs to be applied to the troposphere and the stratosphere separately.

Yes, thank you for these suggestions. For our application, we are not specifically interested in day-to-day variability in the atmosphere, but we agree that including more temporal information in our classification approach would be prudent. We have modified our approach to include seasonal variations by switching from annual mean profiles to seasonal mean profiles. Furthermore, we have also excluded some of the near-surface pressure levels, such that the classification is less affected by near-surface ozone processes. The suggestion to classify the troposphere and stratosphere separately is interesting; unfortunately, there are not enough pressure levels in the stratosphere alone to justify classifying it entirely separately from the troposphere. In any case, we chose this approach because it retains more of the entire profile structure, which we view as an advantage - we want the resulting classes to be influenced by the upper troposphere, the stratosphere, and the interaction between the two.

2.      What is the impact (or limitation) of using annual averages? Ozone concentrations vary widely from summer to winter in both the troposphere and the stratosphere. How different are the clusters if the analysis is applied separately to summer and winter months? Another problem with the annual average is that mid-latitudes are heavily influenced by polar air masses in winter (low tropopause), and by tropical air masses in summer (high tropopause). So the annual average is just an unrealistic homogenization of very different air masses, and does not reflect the typical ozone profiles one might find in any given month or season.

Yes, this is an important point. We have switched to classifying seasonal mean profiles, which allows seasonal variation. The analysis in the paper has been updated to reflect this seasonal variation. Thank you for the suggestion.

The analysis makes no use of observations, and with no evaluation against real-world data we are unable to understand the accuracy of the method. Stauffer et al. (2018) clustered ozone profiles at more than 2 dozen ozonesonde stations worldwide. I realize the authors can't use sparse observations as the basis for this global-scale analysis, but they can certainly evaluate the results against observations. The authors should examine the

observed profiles above the ozonesonde stations that lie within each of the clustered regions. Do the profiles within each region have similar characteristics? If so, then the method is applicable to the real-world; if not, then the usefulness of the method is questionable. What is the result when the observations are then examined by season? Are the observations within each cluster similar to each other in summer, and also in winter? Or does everything break down (see my comment above about seasonal variability in the mid-latitudes).

In this paper, we are focusing on GMM as a model analysis and comparison tool. We agree that applications to observational data would be interesting, but it is beyond the scope of this short technical note. In any case, it is not necessarily obvious how one would carry out such a comparison, given the sparse observational coverage of atmospheric ozone. Specifically, one might try to fit the GMM using observational profiles only and then use those classes to validate and analyze a model run. This has been done for ocean temperature profiles for a specific region in the European Arctic (Thomas and Müller, 2022, https://doi.org/10.1016/j.ocemod.2022.102092). However, given that observational coverage of atmospheric ozone is not especially uniform, it would be difficult to generate a sufficiently general training dataset for GMM. Any resulting GMM would be biased towards the ozone profiles seen at the observing site. We believe we have shown that GMM can, at the very least, be a useful tool for model ozone analysis and comparison.

Other comments:

1. Line 59: When reviewing clustering techniques as applied to ozone profiles, the authors should include Stauffer et al. (2016, 2018).

Thank you for providing some excellent references. We cited them and extended our ozone clustering techniques applied in other studies. The paragraph now reads:

Clustering techniques have already been used in ozone concentration studies for understanding long-term variability. Boleti et al. (2020) have applied a multidimensional clustering technique to understand the long-term trend of ozone. Diab et al. (2004) used a six-cluster analysis which resulted in distinct clusters of "background" and "polluted" with below and above ozone mixing ratios from over 100 ozonesonde profiles launched from a subtropical Southern Hemisphere Additional Ozonesondes (SHADOZ) (Thompson et al., 2003) site, Irene, South Africa. Jensen et al. (2012) performed a cluster analysis named self-organizing maps (SOM) (Kohonen, 2012) on over 900 tropical ozonesonde profiles. Their findings with four-cluster results were similar to Diab et al. (2004). Both studies showed that the seasonal influences of biomass burning and convection dominate ozone variability. Stauffer et al. (2016) documented the influence of meteorological conditions on the shape of the ozone profile from the troposphere to the lower stratosphere by

applying SOM clustering technique to ozonesonde data from specific northern hemisphere midlatitude geographical regions. Later they expanded the study for global ozonesonde sites to show the variation of ozone profiles cluster for various regions and how they vary based on meteorology and chemistry depending on latitudes (Stauffer et al., 2018).

2. Line 61: The perceived methodology and aim of Chang et al. 2017, as stated in the manuscript, is not correct. Chang et al. 2017 are not seeking to cluster similar ozone monitoring sites. Rather they are trying to quantify the regional-scale, long-term trend of ozone while accounting for the spatial distribution of the sites and the correlation between sites. This method accounts for the uneven distribution of sites and prevents any heavily-sampled sub-region from exerting an out-sized influence on the trend

We agree that the citation was not correct. Thank you for pointing that out. We got rid of that part.

3. Line 95-98: I don't understand why the study is limited to 1-1000 hPa. This omits a large section of the globe, i.e., land regions more than 100-200 m above sea level. I realize the method cannot tolerate missing values, but why not conduct the study for profiles in the range of 1-950 hPa; this way, you retain most of the land areas.

 Thank you for the suggestion. We are now focusing on 1-850 hPa to retain more land areas.

4. Line 110: This statement is problematic:"The motivation behind withholding the geographical information is that there is no reason for the vertical ozone structure of the profile to be unique to a given region (Maze et al., 2017)." Using a paper that deals with ocean temperature, the authors seem to suggest that there is no discernable structure in the global ozone distribution and that one region is no different from another. Yet, plenty of observation-based studies identify clear structure in the global ozone distribution that varies with season [Kley et al., 1996; Thouret et al., 1998; Oltmans et al., 1996, 2004; Thompson et al., 2003; Cooper et al., 2007; Gaudel et al., 2020;]. Therefore, certain profile types are more likely to occur in some regions than in other regions. This statement needs to be revised.

We agree that the original statement was unclear. We have revised the text and hope that you find the new statement suitable.

5. Line 173: To say that the tropopause is around 300 hPa is a gross over-simplification. As can be seen in Figure 2, there are plenty of profiles in which the tropopause is around 150 hPa, which is common in the tropics.

We have changed this statement to :

The tropopause height, above which the ozone starts increasing, varies between 300-150 hpa depending on the location of the profiles.

6. Line 175: The statement that ozone increases near the surface is problematic because ozone is plotted in units of mPa. If ozone is plotted in units of ppbv (the typical unit for evaluating air pollution levels in the troposphere), then we would see that the average ozone profile has more ozone in the upper troposphere, especially at mid-and high latitudes (see the ozone profile papers that I cited above). Furthermore, Jaffe and Wigder (2012) is not a sufficient reference because they only discuss ozone at the surface and do not mention the vertical distribution of ozone.

We use ozone partial pressures (mPa) in this study because if we use mixing ratios the ozone profiles would span many orders of magnitude, with surface ozone mixing ratios in the 10s of ppb, and stratospheric ozone mixing ratios reaching a maximum of ppm, and so the profiles would be dominated by the shape in the stratosphere, and it would be very difficult to see what is happening in the troposphere. The choice of mPa shows more clearly the relative structure between profiles in the troposphere. We have amended this paragraph to explicitly highlight the fact that we are discussing ozone concentrations and replaced the reference with that of Monks et al., which provides a more complete review of tropospheric ozone distributions and processes. The paragraph now reads:

Our purpose is to identify coherent patterns within the collection of profiles using unsupervised machine learning. Overall, the profiles reveal relatively high ozone concentrations in the lower and middle stratosphere which peak and then decrease gradually in the upper stratosphere. The tropopause height, above which the ozone concentrations start increasing, varies between 300-150 hPa depending on the location of the profiles. The peak starts decreasing at around 70 hPa and higher altitudes above (Figure 2). In the troposphere, ozone concentrations are fairly constant and then increases towards the surface, in part due the availability of ozone precursors from biomass burning and anthropogenic emissions sources (e.g., Monks et al., 2015).

7. Line 216: Why is the high surface ozone only attributed to biomass burning? This cluster spans the major fossil fuel combustion regions of the northern hemisphere, which are known to drive ozone production across the region.

This has been amended and now refers to ozone precursors from different sources. The sentence now reads:

In the troposphere, ozone concentrations are fairly constant and then increase towards the surface, in part due to the availability of ozone precursors from biomass burning and anthropogenic emissions sources (e.g., Monks et al., 2015).

8. Line 265: The statement that ozone precursor emissions generally increase under SSP5-8.5 isn't really correct, as emissions continue to decrease in developed nations but increase in the developing world. This discussion should also consider the findings of Zanis et al., 2022.

This statement has been expanded upon in the final draft, and we have also included some of the discussion from Zanis et al.. The relevant section now reads:

Here we examine the structure of atmospheric ozone in the 2095-2100 years of the SSP5-8.5 experiment. In this experiment ozone mixing ratios are generally higher throughout much of the troposphere and upper stratosphere. In the troposphere, the drivers of this increase are complex. Under the assumptions of the SSP5-8.5 scenario, global mean emissions of NOx and CO are lower in 2095 than the present day, while global mean emissions of CH4 are higher (Gidden et al., 2019). However, changes to ozone precursor emissions alone do not drive tropospheric ozone changes, which is also affected fby climate change, with increasing tropospheric temperatures changing biogenic volatile organic compounds (BVOC) emissions, the availability of tropospheric water vapor, and stratosphere-to-troposphere transport of ozone, which taken together drive increases to tropospheric ozone concentrations (e.g., Griffiths et al., 2021; Turnock et al., 2021; Zanis et al., 2022). In the stratosphere this increase is simpler to understand. Upper stratospheric ozone increases under all SSPs as ozone depleting substances decrease, but increases more in scenarios which assume larger increases in greenhouse gas emissions due to the resulting CO2-induced cooling of the stratosphere and the impacts this has on gas phase chemistry (e.g., Haigh and Pyle, 1982; Jonsson et al., 2004).

**Minor Revision:**

**Comment to Report 1:**

1. **Line 97: Should this say "aseasonal" or "seasonal"?**

   Thank you for noticing the typo. We fixed it.

**2.      Lines 226-229: This sentence is difficult to follow and might be clearer if it was split into separate sentences for Antarctic versus the NH tropical class widening. Then the discussion of causes of the tropical widening could be expanded a little.**

   Thank you for the suggestion. This sentence now reads as two different sentences. We also extended the tropical broadening paragraph:

The widening trends based on seasonality imply that the tropical broadening in SH is mainly due to the Antarctic ozone hole, which causes the largest radiative cooling effect in the lower stratosphere during DJF (Palmeiro et al., 2014). Increasing black carbon and tropospheric ozone are considered as major forcing for NH tropical class widening on a longer time scale during JJA (Allen et al., 2012). However, these two forcings together have the largest warming effect in the NH extratropics (Hu et al., 2018). Studies showed that the shallow branch (located in the

lowermost stratosphere with upwelling in the tropics and downwelling in the subtropics) of tropical upwelling is much stronger toward the summer hemisphere during DJF than JJA (Palmeiro et al., 2014). The deep branch with upwelling in the upper stratosphere in the tropics and downwelling in the middle and high latitudes also show a similar seasonal cycle with downwelling extended to the polar latitudes in the stratosphere (Seviour et al., 2012; Palmeiro et al., 2014). The differentiation between twp branches are based on different forcing, planetary-scale wave forcing act on the shallow branch, and in the deep branch, the upwelling is associated with GHG increase (Palmeiro et al., 2014). However, the investigation of seasonal change of tropical upwelling in shallow and deep branches is beyond the scope of this study.
.

**3.      Line 246: missing apostrophe on continents and ranges.**

We fixed the error.

**4.      Line 290: Wouldn't changes in BVOC emissions be included in changes to ozone precursor emissions? Maybe clarify if "changes to ozone precursor emissions" means specifically anthropogenic or biomass burning emissions.**

We changed the statement. Since the BVOC term was introduced in this line, we modified the line as below:

However, changes to ozone precursor emissions (including biogenic volatile organic          compounds (BVOC) emissions caused by increasing tropospheric temperature) alone do not drive tropospheric ozone changes, the availability of tropospheric water vapor, and stratosphere-to-troposphere transport of ozone, which taken together drive increases to tropospheric ozone concentrations.

**Comment to Report 2:**

1. **Line 174-176: As I explained in my previous review, the statement that ozone concentrations increase towards the surface is not correct. Figure 2 shows ozone as a partial pressure, which is not a concentration (concentration has units of mass per volume). If you convert the ozone to units of concentration, or ppbv, you will see that the ozone concentrations (or mixing ratios) decrease towards the surface, as shown in Figure 4 of Logan, 1999 (this is a very well-known phenomenon). Also see Figures 5, 6, 16 and 17 of Gaudel et al. (2018). The ozone climatology by Logan (1999) mainly focuses on observed ozone profiles in remote or rural locations, as does your analysis. If one carefully selects ozone profiles from an urban area with very high concentrations in the boundary layer, then one would see an increase of ozone**

**concentrations towards the surface, but this applies to just a very small portion of the Earth's area (see Figure 18 of Gaudel et al. 2018).**

Thank you for the suggestion. We got rid of the part.

**2.      Line 199: Surface ozone is discussed, but the figures only show ozone down to the 850 hPa level, which in most regions of the world, is not the surface. To be accurate please use the expression "lower troposphere" instead of surface. The authors claim that classes 5 and 6 have more ozone in the lower troposphere than classes 3 and 4, but according to Table 1, classes 5 and 6 have less ozone than class 4.**

We reworded the line and changed the word "surface" to "lower troposphere". The line now reads:

Class 4 features higher lower tropospheric ozone and higher variability than class 3. Finally, classes 5 and 6 are northern hemispheric classes with high lower tropospheric ozone concentrations and large variability from the tropopause to the stratosphere. The higher lower tropospheric values result from greater surface pollutants in classes 4, 5, and 6, including the associated ozone precursor emissions, which tend to be concentrated in the Northern Hemisphere due to anthropogenic emissions (Monks et al., 2009, 2015).

**3.      Figure 6: September is misspelled under panel (d).**

Thank you for noticing that. We fixed it.

**4.      Line 239: Here the authors cite Allen et al. 2012, who discuss the impact of long-term climate change (i.e. many decades) on the widening of the tropical belt, and the authors seem to imply that this process is causing class 3 and 4 to be more prominent in JJA, but this is not the case. The seasonal cycle of ozone during the very brief period of 2009-2014 has nothing to do with long-term climate change, and it is simply driven by Earth's normal seasonal dynamics.**

Thank you for the suggestion. We changed the line as below:

Increasing black carbon and tropospheric ozone are considered as major forcing for NH tropical class widening on a longer time scale during JJA (Allen et al., 2012).

**5.      Line 243: Here the discussion of the northern hemisphere ozone hole seems to imply that this phenomenon does not occur in spring. But as shown in the Stratospheric Ozone section of Dunn et al., (2022), the northern ozone hole happens in spring (but it does not form every year).**

Thank you for the suggestion. We modified the statement as below:

This indicates that in our study, the northern hemisphere ozone hole is not especially predominant during these months in seasonal mean. However, Dunn et al. (2022) showed that there are some particular years when the polar ozone hole can happen in NH spring.